



# Present-day radiative effect from radiation-absorbing aerosols in snow

Paolo Tuccella[1], Giovanni Pitari[1], Valentina Colaiuda[1], Gabriele Curci[1]

[1]Departement of Physical and Chemical Sciences, University of L'Aquila, 67010 L'Aquila, Italy

*Correspondence to*: Paolo Tuccella (paolo.tuccella@aquila.infn.it, paolo.tuccella@univaq.it)

**Abstract.** Black carbon (BC), brown carbon (BrC) and soil dust are the most radiation absorbing aerosols (RAA). When RAA are deposited on the snowpack, they lower the snow albedo, increasing the absorption of the solar radiation. The climatic impact associated to snow darkening induced by RAA is highly uncertain. In this work, a 5-years simulation with GEOS-

Chem global chemistry and transport model was performed, in order to calculate the present-day radiative forcing (RF) of RAA in snow. RF was estimated taking simultaneously into account the presence of BC, BrC, and mineral soil dust in snow. Modelled BC and black carbon equivalent (BCE) mixing ratios in snow and the fraction of light absorption due to non-BC compounds ($f_{non-BC}$) were compared with worldwide observations. We showed as BC, BCE and $f_{non-BC}$, obtained from deposition and precipitation fluxes, reproduce the regional variability and order of magnitude of the observations. Global mean

all sky total RAA, BC, BrC and dust snow RF are 0.068, 0.033, 0.0066, and 0.012 W/m$^2$, respectively. At global scale, non-BC compounds account for 40% of RAA snow RF, while anthropogenic RAAs contribute to the forcing for 56%. With regard to non-BC compounds, the largest impact of BrC has been found during summer in the Arctic (+0.13 W/m$^2$). In the middle latitudes of Asia, dust forcing in spring accounts for the 50% (+0.24 W/m$^2$) of the total RAAs RF. Uncertainties in absorbing optical properties, RAA mixing ratio in snow, snow grain dimension, and snow cover fraction result in an overall uncertainty

of -50%/+61%, -57%/+183%, -63%/+112%, and -49%/+77% in BC, BrC, dust and total RAAs snow RF, respectively.





## 1 Introduction

In the last decades, many studies have recognized as the radiation-absorbing aerosols (RAA), in particular black carbon (BC),
brown carbon (BrC) and dust, contribute to the climate warming once deposited on the snow and ice pack (Hansen et al., 2004, 2005, 2007; Flanner et al., 2007, 2009; Bond et al., 2013; Boucher et al., 2013; Lin et al., 2014; Pitari et al., 2015; Skiles et al., 2018). Observations show that the extent of snow coverage is declining, due to global warming (Sturm et al., 2017). RAAs deposition on snowy surfaces results in an enhancement of the absorbed solar radiation in snow, due to the albedo reduction of snow pack. This process increases melting and reduces the snow duration, consequently. As a result, this amplifies the
alteration of runoff timing and magnitude due to the climate warming (e.g., Coppola et al., 2014) with consequences on water resources (e.g.; Painter et al., 2010; Skiles et al., 2012; Wu et al., 2018). Forcing efficacy by RAAs in snow is about three time larger than the one resulting from carbon dioxide ($CO_2$) (Flanner et al., 2007; Bond et al., 2013; Boucher et al., 2013). However, the climatic effect associated to RAAs in snow is still highly uncertain.

BC is emitted by the incomplete combustion of fossil fuel (FF), biofuel (BF) and biomass burning (BB) (Bond et al., 2013),
resulting in high efficiency in absorbing the incoming solar radiation (Bond et al., 2013; Boucher et al., 2013). Current estimates suggest BC as the second most important climate forcing species after carbon dioxide (Gustafsson and Ramanathan, 2016). A large regional variability in BC mixing ratio in snow, ranging over four orders of magnitude, has been found by Warren (2018). As an example, Doherty et al. (2010) reported values of 1-4 ng/g in Greenland ice sheets, 4-10 ng/g in the Arctic Ocean, 8-14 ng/g in Canada and up to 10-60 and 20-60 ng/g in Russian Arctic and Scandinavia, respectively. As for
middle latitudes, BC mixing ratio in snow has found to be in the range of 5-70 ng/g in central North America (Doherty et al., 2014), 20-600 and 30-2000 ng/g in Northwest and Northeast China (Ye et al., 2012; Wang et al., 2013). The lowest BC mixing ratios in snow were founded in Antarctica, being in the order of tenths of ng/g (Warren and Clark, 1990; Grenfell et al., 1994; Zatko and Warren, 2015). Currently, the best estimation of the present-day radiative forcing (RF) by BC in snow and in melting snow-free sea ice is +0.040 W/m² (+0.01/+0.09 W/m²) and +0.012 W/m² (+0.008/+0.017 W/m²) (Bond et al., 2013),
respectively. The resulting change in the surface temperature global average due to RF exerted by BC in snow, ranges from +0.06 to +0.20 K, while equilibrium change in Arctic surface temperature is estimated to be in the range of 0.50-1.6 K (Bond et al. 2013, Flanner et al. 2007).

RAAs in snow are not only constituted by BC, since the presence of both soil dust and absorbing organic aerosol (OA). Measurements across the Arctic and central North America suggested that 20-50% of the sunlight absorption in the snowpack
is due to non-BC RAA particles (Doherty et al., 2010, 2014), while in the Qilian Mountains the snow particulate absorption resulted to be dominated by non-BC compound (around 100%), according to Wang et al. (2013).

BrC is defined as the radiation-absorbing fraction of OA (Andreae et al., 2006; Laskin et al., 2015) and absorbs shortwave radiation with wavelengths less than 400 nm (Lukacs et al., 2007; Alexander et al., 2008; Chen et al., 2010; Arola et al., 2011; Kirchstetter et al., 2012). Sources of such absorbing organic matter are still uncertain; observations show that BrC is mainly



produced by BF combustion, BB, and ageing of secondary organic aerosol (SOA) (Bones et al., 2010; Hecobian et al., 2010; Arola et al., 2011; Updyke et al., 2012; Lambe et al., 2013; Laskin et al., 2015; Guang et al., 2016), while other sources are represented by some aqueous-phase chemical reactions in clouds (Zhang et al., 2017). BrC treatment is poor in current atmospheric models due to lack of mass and absorption observations. For this reason, OA is commonly treated as scattering and only few studies have investigated climatic impact of OA as BrC (Wang et al., 2014; Lin et al., 2014; Saleh et al., 2015;

Jo et al., 2016; Wang et al., 2018; Brown et al., 2018; Tuccella et al., 2020). Last IPCC report did not treat the absorption of BrC in snow. To the best of our knowledge, the global RF due to the change of OA in snow from preindustrial time has been calculated by Lin et al. (2014), only, who reported estimated values ranging from +0.0011 to +0.0031 W/m$^2$.

Soil dust are emitted from arid and semi-arid regions (Choobari et al., 2013), and from soils disturbed by anthropogenic activities (Tegen et al., 2004; Ginoux et al., 2012). The largest amount of dust in snow was found downwind from large sources

(Skiles et al., 2018). Soil dust is much less absorbing than BC, but it may dominate the RF in snow when is present in very high concentrations. Although dust emissions have increased (Mahowald et al., 2010), the last IPCC report did not asses the radiative forcing of dust in snow.

Some studies have shown that the presence in snow of non-BC absorbing impurities reduces the influence of BC, especially in regions located downwind from large dust sources (Bond et al., 2013). Flanner et al. (2009) found that the dust deposition

on the snowpack decreases the BC RF by 25%. According to Bond et al. (2013), the role of dust in reducing BC RF ranges from 10% to 40%. Dang et al. (2017) have reported regional-averaged albedo reductions due to RAAs of 0.009, 0.012 and 0.077 for new snow in the Arctic, North America, and China, respectively. Moreover, in the same regions, the albedo reductions caused by BC only are 0.005, 0.005, and 0.031. The same authors have also estimated an increase of 20-40% of albedo perturbation in snow not containing non-BC RAAs. On the other hand, the presence of different light-absorbing

impurities may also impact the BrC RF. Beres et al. (2020) found that the BrC deposition onto pure snow resulted in a more than twice local instantaneous RF, with respect to a case with dark snowpack. These results highlight the importance of simultaneously taking into account the concentrations in snow of BC, BrC, and mineral soil dust.

RF by snow RAAs is also affected by other uncertainties related to emissions, snow ageing, scavenging of the impurities in melting snow, parameterization for snow cover fraction (SCF), and absorption optical properties. The overall error associated

to BC RF due to these single uncertainties is -73%/+117% (Flanner et al. 2007; Flanner et al. 2009; Bond et al., 2013). To the best of our knowledge, there are no studies which quantify all these uncertainties for BrC and soil dust at global scale.

In this study, we have performed a multi-year simulation of the RAA mass concentration with the global chemical and transport model GEOS-Chem (Bey et al., 2001). Starting from the GEOS-Chem output, we have diagnosed the mass mixing ratios of RAAs in snow and, subsequently, calculated the associated RF, taking into account the simultaneous presence of BC, BrC,

and dust in the snow. RAAs mass and their optical properties have been calculated using the most recent updates in terms of ageing, size distribution and absorption optical properties, inferred from observational constraints consequent to our previous



work (Tuccella et al., 2020). We also explored the sensitivity of RF from RAAs in snow to the assumptions on their optical properties, mixing ratio uncertainty, snow ageing, and SCF.

## 2 Methods

### 2.1 GEOS-Chem model

Aerosol mass concentration was simulated with the GEOS-Chem global chemical and transport model, v11-01 (Bey et al., 2001, with updates documented in www.geos-chem.org). Five years from 2010 to 2014 were simulated by the model, with a grid resolution of $4° \times 5°$ and 47 vertical levels up to 0.01 hPa. Herein, GEOS-Chem was driven by Modern Era Retrospective-analysis for Research and Application version 2 (MERRA2) assimilated meteorological data from the Global Modelling and Assimilation Office Goddard Earth Observing System (Rienecker et al., 2011).

We carried out a full aerosol-oxidant chemistry GEOS-Chem simulation, including: sulphate, nitrate, ammonium, black carbon, organic aerosol, soil dust, and sea salt. BC and primary organic aerosols (POA) were calculated according to Park et al. (2003). SOAs were parameterized by using the scheme of Pye et al. (2010), while dry deposition was simulated with a resistance-in-series model (Zhang et al., 2001). Finally, the wet deposition processes were parameterized through the scheme from Liu et al. (2001) and included the both the below-cloud washout from large-scale and convective precipitation, and in cloud removal for stratiform clouds and convective updrafts.

Primary anthropogenic emissions of BC and POA were taken from Bond et al. (2007) inventory. Global anthropogenic emissions of CO, NOx, and SOx were taken from Emissions Database for Global Atmospheric Research (EDGAR) v4.2 ($1°$ $\times 1°$). The volatile organic carbon (VOC) emissions were from the REanalysis of the TROposhperic chemical composition (RETRO) ($0.5° \times 0.5°$) inventory. Regional inventories were used to replace EDGAR and RETRO, such as: EMEP (50 km $\times$ 50 km) for Europe, NEI2011 (12 km $\times$ 12 km) for the United States, BRAVO ($0.1° \times 0.1°$) for Mexico, CAC ($0.1° \times 0.1°$) for Canada, and Streets et al. (2006) data (1 km $\times$ 1 km) for Asia.

The biomass burning emissions of BC and POA followed the year-specific daily mean GFED4s (Global Fire Emissions Database with small fires) inventory (van der Werf et al., 2010; Giglio et al., 2013), while the biogenic emissions were calculated interactively within GEOS-Chem, with the Model of Emissions of Gases and Aerosols from Nature (MEGAN) (Guenther et al., 2006). Dust emission flux was simulated through the Dust Entrainment And Deposition (DEAD) scheme (Zender et al., 2003), and the dust source function taken from the Goddard Chemistry Aerosol Radiation and Transport (GOCART) model (Ginoux et al., 2001; Chin et al., 2004).

We used a modified version of GEOS-Chem which included a specific treatment for RAAs, following our previous work (Tuccella et al., 2020). BC emissions and ageing were considered as source-dependent as in Wang et al. (2014, 2018) and hydrophobic and hydrophilic BC were tracked for FF, BF, and BB sources. According to Wang et al. (2014), 80% of BC from



FF sources was emitted as hydrophobic and converted to hydrophilic with an ageing rate depending on sulphate dioxide and hydroxyl radical levels in the atmosphere (Liu et al., 2011). By contrast, BC from BF and BB sources was assumed to be emitted as 70% hydrophilic and 30% as hydrophobic with an ageing e-folding time from hydrophobic to hydrophilic of 4 hours.

As in Wang et al. (2014), BrC emissions were inferred from POA emissions, assuming 50% and 25% of POA from BF and BB emission as primary BrC. Moreover, we have assumed that half of emitted BrC is hydrophobic, with a conversion time of hydrophobic BrC to hydrophilic of 1.15 days.

BrC SOA is produced by many sources. Some studies showed as absorbing SOA is contained in aromatic compounds (Lambe et al., 2013), however, it may also derive from browning of some anthropogenic and biogenic SOA by reaction with ammonium, from photooxidation of α-pinene and toluene in the presence of $NO_x$ and from the reaction of limonene with $O_3$ (Bones et al., 2010; Updyke et al., 2012). Other sources of BrC are aliphatic compounds (Laskin et al., 2015; Guang-Ming et al., 2016) and aqueous-phase chemical reactions in clouds (Zhang et al., 2017). The fraction of absorbing SOA in atmosphere is not well constrained, thus we assumed that all SOAs simulated by GEOS-Chem are BrC, following Lin et al. (2014). These, include: compounds from photooxidation of light aromatics, aerosol formed from photooxidation, ozonolysis, nitrate radical oxidation of monoterpenes and sesquiterpenes and products of isoprene oxidation (Pye et al., 2010).

Dust mass was simulated with four dimensional bins, with the following diameter boundaries: 0.2–2.0, 2.0–3.6, 3.6–6.0 and 6.0–12.0 μm. Emitted dust was distributed among these bins following Kok (2011). Dust emission was adjusted to give a global mean burden of 20 Tg which is the central estimate reported by Kok et al. (2017), calculated from observational constraints. Further details are provided in Tuccella et al. (2020).

## 2.2 Snow albedo perturbation

Several methods for the calculation of snow albedo reduction caused by BC deposition have been reported in the scientific literature. As an example, a simple approach was proposed by Hansen and Nazarenko (2004), where the deposition fields at each surface grid box were scaled with the albedo perturbation values (Hansen et al. 2005, 2007; Pitari et al. 2015). This method is affected by large uncertainties as some factors influencing the snow albedo (i.e. the snow aging and internal/external mixing of BC) are not considered (Hansen et al., 2004; Pitari et al., 2015). One of the most advanced method consists in applying a radiative transfer model in the snow, coupled to a chemical and transport model and a snowpack scheme including snow ageing and a prognostic treatment for snow impurity content (Flanner et al., 2007, 2009).

Herein, we used an approach of intermediate complexity. The snow albedo reduction by RAAs was calculated through the parameterization of Dang et al. (2015), which is based on the Mie theory for spherical particles (Mie, 1908) and assumes that snow impurities are externally mixed with snow grains. In this scheme, the snow albedo reduction by BC is parameterized for three broad bands: all-wave, visible (VIS) and near-infrared (NIR), by using a quadratic or cubic polynomial in BC mixing





ratio, whose coefficients are themselves quadratic in snow grain size ($R_e$). It should be noted that the parameterization of Dang et al. (2015) has been formulated assuming a size distribution and a complex refractive index for BC, however, it may be used

for BC particles with different size and refractive index, by scaling the mass mixing ratio with the ratio between the mass absorption coefficient (MAC), based on own assumptions, and the one used in Dang et al. (2015). MAC of given aerosol species at a given wavelength λ is defined as:

$$MAC_\lambda = \frac{3Q_{ext,\lambda}(1-\omega_\lambda)}{4\rho R_{eff}} \tag{1}$$

where $Q_{ext}$ is the extinction efficiency, $\omega$ the single scattering albedo, $R_{eff}$ is the particle effective radius, and $\rho$ the particle

density. In our work, the contribution to snow albedo reduction from soil dust and BrC was taken into account through the concept of black carbon equivalent (BCE) (Grenfell et al., 2011), following Ward et al. (2018):

$$BCE = BC + \sum_{i=1}^{4} \frac{MAC_{Dust,i}}{MAC_{BC}}[Dust_i] + \frac{MAC_{BrC_{BF}}}{MAC_{BC}}[BrC_{BF}] + \frac{MAC_{BrC_{BB}}}{MAC_{BC}}[BrC_{BB}] + \frac{MAC_{BrC_{SOA}}}{MAC_{BC}}[BrC_{SOA}] \tag{2}$$

being *[BC]* the BC snow mixing ratio, *[Dusti]* the mixing ratio of dust in the dimensional bin *i*-th bin and *[BrCBF]*, *[BrCBB]*, *[BrCSOA]* the BrC mixing ratios from BF, BB, and SOA sources, respectively. Once the BCE has been calculated, the snow

albedo reduction from all absorbing impurities may be estimated with the parameterization for BC proposed by Dang et al. (2015). It should be noted as the MACs appearing in the latter equation are spectrally averaged over an incident solar spectrum, which is characteristic of summer high-latitude conditions. The limits of integration are those of the band of interest. MACs adopted in this work are discussed in the next subsections.

Dang et al. (2015) also provided a scheme for the albedo reduction of snow containing dust particles, gathered through an

estimation of the BCE. However, this parameterization has not been used in our work for the following reasons: first the scheme is based on assumptions about the refractive index and size distribution. In particular, a single log-normal mode for long-range-transported dust was adopted, while, in our model, dust size distribution was evolving in time and was not log-normal. Secondly, our simulations included the coarsest dust particles funded near sources, that could not be well represented by the size distribution adopted by Dang et al. (2015).

Snow albedo is sensitive to grain size, which depends on snow aging SA processes (Flanner et al., 2006; Flanner et al., 2007). Old snow presents larger $R_e$ then new fresh snow and the albedo variation for a given snow concentration of absorbing impurities is greater in larger grained snow, than in smaller ones. A large number of studies considered a constant $R_e$, while few works only had used a physical model to calculate the $R_e$ growth (Bond et al., 2013). Uncertainties in $R_e$ results in an error of -42%/+58% in RF estimation (Flanner et al., 2007), for this reason, in order to take into account $R_e$ seasonal and geographical

variability, we used the exponential relationship proposed by Zhao et al. (2013) to calculate $R_e$, starting from the snow albedo inferred from MERRA2 reanalysis. It should be noted that this is a rough approximation, since the relationship of Zhao et al. (2013) was based on the snow reflectance measured near 1030 nm, while the snow broadband albedo for the calculation is





used in our work/experiment. Moreover, the relationship derived from the snow fields of Heihe River Basin (China) and could not be consistent in other regions.

Finally, impurity mixing ratios in snow were calculated as the ratio between the deposition fluxes of BC, dust, and BrC simulated by GEOS-Chem, and MERRA2 total precipitation flux (Kopacz et al. 2011; He et al., 2014). It should be noted as the impurity content in snow is not only determined by the accumulation rate from deposition processes, but it is also a function of the scavenged fraction of impurities by melting snow. According to Flanner et al. (2007), uncertainties in the scavenged fraction produce an error in BC RF estimation ranging between -31% and +8%.

**2.3 Radiation-absorbing aerosol optical properties**

In this work, the same set of optical properties for RAAs employed by Tuccella et al. (2020) has been used. The size distribution for BC is sources-dependent, according to Wang et al. (2018). The geometrical median radius was fixed to 30 and 70 nm for FF and BF/BB black carbon, while the standard deviations were 1.4 and 1.6, respectively. Following the recommendation of Bond and Bergstrom (2006), the applied refractive index was 1.95-0.79i. The BC density was assumed to be 1.8 g/cm$^3$. Using

the Mie theory (Mie, 1908), the resulting MACs at 550 nm were 6.3 and 6.2 m$^2$/g for FF and BF/BB BC, respectively.

The geometrical median radius, standard deviation and density of BrC are 90 nm, 1.6 and 1.3 g/cm$^3$, respectively (Wang et al., 2018). The real part of BrC refractive index is the same of white OA and was taken from the Optical Properties of Aerosol and Cloud (OPAC) database (Hess et al., 1998). The imaginary part has been inferred starting from the MAC of BF and BB absorbing OA reported by Wang et al. (2018). MAC$_{OA}$ at 440 nm used for the BF was 0.76 m$^2$/g. For freshly emitted

(hydrophobic) BB, MAC$_{OA}$ at 440 nm was 0.77 m$^2$/g. According to Wang et al. (2018), we applied a reduced MAC$_{OA}$ for aged (hydrophilic) OA of 0.23 m$^2$/g. We have used Two different MAC$_{OA}$ for freshly emitted and aged BB OA have been used, with the aim of taking into account the blanching process of aged plumes, due to the photochemical ageing of wildfires plumes (Forrister et al., 2015).

Following again Wang et al. (2018), MAC$_{OA}$ was translated to MAC$_{BrC}$ using the relationship:

$$MAC_{OA} * Mass_{OA} = MAC_{BrC} * f * MAC_{OA} \tag{3}$$

where $f$ is the assumed BrC fraction of OA mass. In our work, BrC was set to 50% and 25% for BF and BB OA, respectively. The resulting MAC$_{BrC}$ values at 440 nm were 1.56, 3.08, and 0.92 m$^2$/g for BF, fresh and aged BB, respectively. BrC from secondary sources was assumed to have the same size distribution of primary BrC; and a MAC$_{BrC}$ of 0.3 m$^2$/g at 440 nm has been assigned to this species, as reported in Wang et al. (2014).

Dust optical properties were also inferred as in our previous work (Tuccella et al., 2020). GEOS-Chem simulates the dust size distribution over four dimensional bins (see Section 2.1), however, the finer bin was split into four bins, for optical calculations, as in Ridley et al. (2012). Dust bins were assumed to be allocated over a gamma distribution, however, their contribution to the optical calculations is given in limited size ranges. Soil dust density was assumed to be 2.5 g/cm$^3$. The refractive index for





mineral dust particles was derived from the datased provided by Petzold et al. (2009). Under these assumptions, Mie calculation
indicates a MAC at 550 nm of 0.057 and 0.048 m$^2$/g for size range of 0.36–0.6 and 4.4–6.0 µm, which are representative of
the dust particles laying far and close to the sources, respectively.

### 2.4 Numerical experiments

A series of numerical experiments has been carried out, in order to study the sensitivity of RAAs snow RF, due to the i)
simultaneous presence of several light-absorbing impurities; ii) their absorbing optical properties; iii) their emissions and
mixing ratios in snow; iv) snow grain size and v) snow coverage fraction. The list of our experiments is reported in Table 1.

The first simulation performed represented our reference case (CTRL), where a simultaneous presence of BC, BrC is
considered in the snow. MACs for RAAs discussed in Subsection 2.2 were spectrally averaged in VIS and NIR bands over an
incident solar spectrum characteristic of summer high-latitude conditions. The absorption enhancement (E$_{abs}$) of BC due to the
"lensing effect" (Lesins et al., 2002), caused by coating of non-absorbing material, was taken into account increasing the MAC
of aged (hydrophilic), by 1.5 a factor, as recommended by Bond et al. (2013). We considered the CTRL simulation as the
"central" (or "mid") absorption scenario. MACs averaged in the VIS are listed in Table 2, those adopted for NIR are reported
in Table S1

In the second simulation, the RF of each single species at time (OSPT) in snow was calculated. The experiment main purpose
was to test how much the presence of more RAAs in snow affects the RF of a single species.

The next performed six perturbed experiments were aimed at evaluating the sensitivity of RF to the assumed absorption aerosol
properties. For these experiments, we have defined a "high" and a "low" absorption scenario for absorbing aerosol species.
Estimation of coated E$_{abs}$ is highly uncertain; recent studies have found values above the most accepted amount of 1.5. Tuccella
et al. (2020) reported values in the range of 1.7–1.9 for BC coated by non-absorbing shell. According to the same authors, E$_{abs}$
is 2.8-3.4 for BC coated by an absorbing shell (brown carbon). Curci et al. (2019) also estimated similar values. On the other
hand, E$_{abs}$ values could be lower than 1.5, as an example, Cappa et al. (2012) have observed very low values for E$_{abs}$ (about
1.1). As a consequence, we applied E$_{abs}$ of 1.9 and 1.1 for BC "high" (BC-H) and "low" (BC-L) absorption scenario,
respectively.

MAC adopted for BrC has been optimized with regional observations in the United States (Wang et al., 2018), therefore,
assumed values/estimation may be not consistent worldwide. MAC used for aged BB BrC were deduced from limited dataset
(Wang et al., 2016), not able to provide the most required constraints on the photochemical whitening processes (Wang et al.,
2018). Moreover, it is not clear if the blanching processes affects the BF BrC, for this reasontwo extreme conditions for BrC
absorption have been tested. In detail, a no whitening process for aged BB BrC was assumed in "high" absorption scenario
(BrC-H), while whitening of BF BrC is considered in "low" absorption scenario (BrC-L). In the latter one, we assumed that
MAC of aged biofuel BrC is reduced by a factor of 2.34 with respect to CTRL run,





Soil dust absorption and its climatic impact strongly depends on the imaginary part of the refractive index (Pitari et al., 2015; Tuccella et al., 2020), which is determined by the mineral composition of soil in the source region. Herein, we explore the sensitivity of dust absorption in the snow due to the refractive index. In particular, the dataset of Wagner et al. (2012) was exploited for the "high" absorption simulation (DUST-H), while the refractive index from Sinyuk et al. (2003) was used in the "low" absorption experiment (DUST-L). As shown in Table 2, considered dust size ranges of MACs are larger up to 3.5 times

in DUST-H with respect to CTRL. By contrast, MACs of DUST-L are lower up to a 1.6 factor with respect to the reference simulation.

The next two experiments were aimed at evaluating the sensitivity of RAAs snow RF to their concentration in snowpack. The mixing ratio of absorbing impurities in the snow depends on many factors, such as emissions, deposition and precipitation rates, impurities solubility and snow melting (Flanner et al. 2007; Bond et al., 2013). Previous studies had shown that BC

absorption is underestimated in GEOS-Chem and this was partly related to the uncertainties in both anthropogenic and biomass burning emission inventories (Wang et al., 2014; Jo et al., 2016; Tuccella et al., 2020). Data analysis and modelling had revealed that current inventories underestimate the emissions from shipping and petroleum extraction in the Arctic (Tuccella et al., 2017; Law et al., 2017). On the other hand, dust emission is also uncertain. As an example, Kok et al. (2017) reported a range of values within 1000-2700 Tg/yr, for global $PM_{10}$ dust, based on observational costraints. Furthermore, another

uncertainty factor affecting our simulations, is associated to the fraction of POA emission that we assumed as BrC. Varying the emitted fractions by a factor 1.5, the uncertainty in BrC absorption is about 30% and 40% for BrC, from BF and BB sources, respectively (Jo et al. 2016; Tuccella et al., 2020). In order to evaluate the impact of all these uncertainties in snow RF calculation, we perturbed the RAA mixing ratios in snowpack by doubling (BCE-H) and halving (BCE-L) the BCE.

The last two experiments were performed to test the sensitivity of RAA snow RF to $R_e$ and SCF. As explained in the Subsection

2.2, we have estimated $R_e$ with a rough method starting from the broad band snow albedo derived from MERRA2 data. In order to explore the impact of uncertainties in RAA snow RF related to $R_e$ estimation, we have multiplied ($R_e$-H) and dived ($R_e$-L) the snow grain radius by a factor 2.

SCF controls the area where RAA snow RF acts. In models, SCF is usually calculated using the snow depth (Flanner et al., 2007; Bond et al., 2013). According to Flanner et al. (2007), SCF may differs up to a factor of 2 among different

parameterizations. This results in an uncertainty of -17%/+8% in BC snow RF (Bond et al., 2013). In MERRA2, SCF fraction is parameterized as a function of the snow water equivalent (SWE) (Rienecker et al., 2011). In particular, SCF = min (1 , SWE/WEMIN), where WEMIN assumes a value of 26 kg/m², representing the minimum SWE to consider over a land grid tile completely covered by snow. In order to evaluate how SCF uncertainty impacts the RAA snow RF, SWE has been multiplied (SCF-L) and divided (SCF-H) by a factor of 1.5.



## 2.5 Radiative transfer model

RAA snow RF due to snow albedo reduction was calculated with the Rapid Radiative Transfer Model for General Circulation Models (RRTMG) (Iacono et al., 2008). RRTMG was interfaced with GEOS-Chem output as described by Jo et al. (2016) and Tuccella et al. (2020). Monthly temperature, water vapour, cloud liquid and ice water path, cloud fraction and surface albedo for direct and diffuse radiation in VIS and NIR bands were taken from MERRA2 reanalysis. Tropospheric and stratospheric ozone were derived from the GEOS-Chem output. Cloud droplet and ice crystal effective radii were from monthly MODIS-Aqua cloud products. For all-sky conditions, vertical overlap of cloudy layers was handled using the Monte Carlo independent column approximation (McICA) (Pincus et al., 2003). Long-lived gases were fixed to climatological mixing ratios. Aerosol optical depth, single scattering albedo and asymmetry parameter used for the atmospheric radiative transfer were calculated using the post-processing tool FlexAOD (http://pumpkin.aquila.infn.it/flexaod/) (Curci et al., 2015; Curci et al., 2019). Finally, aerosol particles suspended in the atmosphere were assumed to be in external mixing.

In summary, BC, BrC, and dust mixing ratio in snow have been estimated starting from the deposition fluxes calculated by GEOS-Chem and total precipitation flux provided by MERRA2 data. BCE was calculated through Equation 2. After having diagnosed the BCE, the perturbation to VIS and NIR snow albedo was calculated according to Dang et al. (2015), as described in Subsection 2.2. The perturbation was applied to MERRA2 surface albedo in the area covered by snow. Finally, the perturbed albedo was used as an input parameter in the radiative transfer model.


## 3 Results

### 3.1 Model evaluation

BC and BCE mixing ratio in snow diagnosed from GEOS-Chem deposition fields have been compared with the worldwide

observations reported in different works. For Arctic and North America, we used the data from Doherty et al. (2010, 2014).
For China, we inferred the measurements from Ye et al. (2012) and Wang et al. (2013), while for Himalya and Tibet Plauteau
the observations were taken from Kopacz et al. (2011). Data for Antarctica were provided from Grenfell et al. (1994), Chylek
et al. (1987), Warren and Clarke (1990), and Zatko and Warren (2015).

In Figure 1, the scatter plot obtained from the comparison between observed and simulated BC mixing ratios in snow surface

layer is reported. Moreover, results from the comparison are summarized in Table 3. Model simulations were compared to the
observed regional median and standard deviation, provided in reference papers. Where these information were not available,
the analysis was carried out by considering the observed regional mean, together with the minimum and maximum values
measured in the region of interest. When measurements corresponded to a specific period falling within the time interval of
our simulations (2010-2014), we have compared model results for the same time frame, otherwise, we the 5-year average over

the same months has been used.

The $R^2$ coefficient between observed and calculated BC mixing ratio was 0.84. This result indicates as the regional variability
of BC in snow, spanning over 5 order of magnitudes, is reproduced by the model. Most part of modelled values (80%) was
found to drop within a factor of 2 from the observations. This last outcome reflects the model skill in reproducing both the
long-range BC transport and the impact of local major sources on the regional snow darkening. In general, the median bias of

the modelled BC was -13%.

Concerning the Arctic, a significant bias was found in Greenland, during summer: in this area, modelled BC is about 7 times
larger than the observed median mixing ratio. As discussed in Doherty et al. (2010), BC concentration in Greenland in
summertime is larger in the melting layer (around 10 cm depth), with respect to the surface, as BC is left on the surface by the
melted water. As this process is missing in calculation schemes, it likely explains the obtained model bias in this case. In

Eastern Russia, the modelled regional median was underestimated by a factor of 3. This bias could be related to both
measurement uncertainties and model prediction. In fact, in some sites of Eastern Russia, BC measurements were affected by
local soil dust and assumptions about the absorption Angstrom exponent (AAE), done for the BC and non-BC components
derivation in snow samples (Doherty et al., 2010). Moreover, some samples were not representative of the regional background,
because they are affected by local sources from villages (domestic wood-burning) and coal-fired power plants (Doherty et al.,

2010), which could not be properly resolved by GEOS-Chem, given the raw resolution used in this work.

As for Antarctica, the model reproduced observed BC amount in snowpack and sea ice, except for Simple Dome station, where
the mixing ratio is underpredicted by one order of magnitude. A similar bias was reported by Flanner et al. (2007), even if the
authors had used a most advanced scheme to calculate BC content in snow. Moreover, the BC concentration measurement at



Simple Dome (Chylek et al., 1987) is old (1982-1985) and may be not representative of the present-day Antarctic BC in snow,
being much larger than more recent observations (0.20-0.60 ng/g), shown in Table 3.

The highest BC mixing ratios in snow are observed in China: the model reproduced the average magnitude of BC in snow
detected in several regions of the country, with the exception of the industrial Northeast district. In this region, predicted BC
concentrations are underestimated by a factor of 3, if compared to the typical values in the range of 1000-2000 ng/g in snow
surface (Wang et al., 2013). This area, where measurements reported by Wang et al. (2013) were collected (Table 3), hosts the
highest number of industrial activities. As a consequence, the model negative bias found for this region may be another effect
attributable to the adopted model resolution.

In Figure 2, the scatter plot resulting from the comparison between observed and simulated BCE (which is defined in Equation
2) mixing ratio in snow surface layer is shown. Numerical results are also reported in Table 3. The $R^2$ coefficient between
observed and calculated BCE mixing ratio was 0.60. As for BC, 80% of the modelled values were within a factor of 2 from
the corresponding observations, resulting in a correct simulation of BCE regional variability. The median bias between
observed and of modelled BCE was -21%: the highest BCE bias was found in two regions of China, in particular, BCE was
underpredicted by a factor of 3 in Qilian Mountains and Northeast Industrial region. By contrast, BCE resulted to be
overpredicted by a factor of 3-4 in Greenland during summer months. In this case, the analysis of the light fraction absorption
due to non-BC compounds ($f_{non-BC}$) revealed other aspects of the model skill in reproducing the RAA in snow.

In Figure 3 a comparison between calculated and observed $f_{non-BC}$. $f_{non-BC}$ is shown proposed, where a $R^2$ of 0.44. is reported,
as well as a 90% of the modelled values being within a factor of 2 from observations. Generally, $f_{non-BC}$ was underestimated
by the model, with a median bias of -17%. The highest bias was found for seasonal snow in North America. Furthermore, a
factor 2 underestimation of $f_{non-BC}$ has been found in Intramountain Northwest (Rocky Mountain), Northern U.S. Plains (North
Dakota), and Canada. In this case, it should be underlined as most part sites in Intramountain Northwest and Northern U.S.
Plains are characterized by thin and patchy snow, and, therefore, affected by soil dust emitted by local sources (Doherty et al.,
2014). This feature is not resolved by the model, due to the raw resolution employed. Soil dust particles also contributes to
absorption in Canadian snowy sites, implying a long range transport (Doherty et al., 2014) which is not simulated by GEOS-
Chem, being some local sources could missing in the model.

$f_{non-BC}$ was underestimated by a factor of 2 in Greenland during spring. In this period, snow RAAs in Greenland is usually
dominated by biomass burning (Doherty et al., 2010), therefore, the $f_{non-BC}$ bias is likely attributable to the BrC treatment in
the model. Moreover, the simulated underestimation could be associated to the POA emissions from BB, as well as the
absorption optical properties, assumed to scale aged BB BrC mass in BCE calculation. Finally, in western Russia, the
underestimation of BC resulted in a highly biased $f_{non-BC}$ (factor 1.5).





In summary, model evaluation through worldwide observations showed a model skill in reproducing BC and BCE mixing ratios in snow. Obtained biases were mainly linked to the emissions of BC, POA, and dust emissions, while error in BCE and $f_{\text{non-BC}}$ simulations is likely related to the assumptions about the RAA optical properties. A source of uncertainty in model evaluation could be represented by measurement errors. For example, according to Doherty et al. (2010), margins of error in BC and BCE measurements are related to both instrumental errors and assumptions made on the aerosol absorbing optical

properties, resulting in a total uncertainty estimation up to ±50%.

### 3.2 Present-day global radiative forcing

In Figure 4, the spatial pattern of the annual surface RF present-day average (2010-2014) from all RAAs, BC, BrC, and dust in snow is shown, as estimated from CTRL run and in all-sky conditions. In Table 4 a summary of the RF estimated from all numerical experiments discussed in Section 2.4 is given. In the following discussion, anthropogenic RAAs were considered

as particles emitted by FF and BF sources and SOA formed by light photooxidation of aromatic compounds. BC and BrC from BB and biogenic SOA were considered as natural. Soil dust was also considered as a natural aerosol, even though about 20-25% of the total present-day emissions have been attributed to human activity (Ginoux et al., 2012); these anthropogenic dust sources were not taken into account in this analysis.

Global average RF associated to all RAA was +0.068 W/m². The largest values were founded in Northeastern China and Tibet

Plateau. As expected, global RAA snow RF was dominated by BC RF, resulting in +0.033 W/m². This value is about 18% lower than the best estimation of 0.040 W/m², reported by Bond et al. (2013). In Figure S1, RF of BC, divided by source, is shown. About 80% (+0.025 W/m²) of BC snow RF was due to the anthropogenic sources. FF BC in snow RF acts everywhere, especially in Southeastern Canada, Eastern Greenland, Northeastern China, and Tibet Plateau. BF BC shew an impact in Eastern Europe, Northeastern China and Tibet Plateau. Eventually, BC from biomass burning occured in Siberia and high

latitudes, as a consequence of the boreal fires.

In our model, soil dust is the second light absorber in snow, having an average RF of +0.012 W/m² which was about 3 times lower than the one due to BC. RF of dust in snow was relevant in the Asian regions, especially downwind the deserts and Tibet Plateau, where values up to +1.7 W/m² are simulated. In some regions of Kazakhstan, Mongolia, Manchuria, Tibet Plateau, Pakistan, and Afghanistan, dust snow RF is on average 2-3 times larger than the one exerted by BC. In Mongolia, dust RF is

up to 4 times larger than BC.

Estimation of snow RF for BrC was +0.0066 W/m², about 5 times lower than the one calculated for BC. Lin et al. (2014) had reported values of 0.0011-0.0031 W/m² for RF of BrC in snow, as a result of an OA emission change of 60 Tg/yr, since preindustrial time. Starting from this RF, normalized to the emission change, the present-day RF may be estimated by using the current OA emission (124 Tg/yr) used in Lin et al. (2014). According to this scaling, the resulting RF was 0.0020-0.0055

W/m², therefore our estimation is found to be above previous upper bounds. In Figure S2, the snow RF of BrC, divided by source, is displayed. BF BrC radiative effect is relevant in Northeastern China and Tibet Plateau, while BB BrC dominates at





high latitudes. Based on our model, BF and BB contribution to annual BrC snow RF was about 38% (+0.0025 W/m²) and 47% (+0.0031 W/m²), respectively. SOA accounts for 15% (+0.0010 W/m²) of OA absorption and its effect is limited at high latitudes in late spring and summer.

It should be noted as the sum of BC, BrC, and dust snow RF is lower than the forcing of all RAAs. Albedo reduction does not increase linearly with the addition of RAAs, because the light penetrating in snowpack decreases as RAA concentration increases (Dang et al., 2017). As a consequence of this non-linearity, it is not straightforward to calculate the relative contribution of each absorbing impurity to total RF. An estimation may be given through the sequential factor separation analysis (Schär and Kröner, 2017); according to this method, non-BC compounds account for about 40% of the absorption in 390 snow. In addition, we found that carbonaceous aerosols (BC+BrC) control about 75% (+0.046 W/m²) of snow RF exerted by RAAs. The contribution of anthropogenic emissions to RAA absorption in snow is around 56% (+0.031 W/m²), meaning that slightly less than half of RAA snow forcing is due to natural sources.

### 3.3 Present-day regional and seasonal radiative forcing

In this section a further investigation on the regional and seasonal dependence of RAAs snow RF is proposed. To this aim, the 395 globe has been divided into five regions, as defined in Table 5. In Figure 5, RF values from RAAs in snow at regional scale are represented, as a function of season. The same figure also reports the relative contribution of each species to total RF.

In Arctic, total RAA snow RF was +0.83 and +0.59 W/m² in spring and summer, respectively, being about 40% of this forcing attributable to non-BC compounds. According to our model, BrC contributed for 14% to total absorption in spring and reached the maximum values (+0.13 W/m²) in summer, where the 32% of overall RAA RF is concentrated. BrC RF is higher in spring, 400 especially on the snow land of Siberia and European high latitudes (Figure S3). This forcing is linked to both BF and BB sources. During summer, BrC RF is dominated by BB, as SOA accounted for about 13% (+0.017 W/m²) of BrC forcing. By contrast, soil dust snow RF has been found to be maximum in spring (+0.12 W/m²) accounting for 24% of the total forcing, whilst its radiative impact in snow is negligible in summer. Dust RF in Arctic is limited to the spring months during Arctic haze transport period, reflecting the absence of significant sources at high latitudes. Arctic dust absorption in springtime is 405 important in Siberia, while minor impact has been found in North America and sea ice (Figure S4). Moreover, about 60% of RAA snow RF in the Arctic was due to anthropogenic sources, in spring. In summer, anthropogenic contribution dropped to 30%, as RF is dominated by BB. In fall, calculated RAA snow radiative effect was +0.15 W/m², being 30% of this forcing attributable to non-BC aerosols, while 65% is determined by anthropogenic sources.

The lowest values of RAA snow RF in the middle latitudes have been found in North America, where the total RAA forcing 410 resulted in 0.15 and 0.17 W/m² in winter and spring, respectively. In North-America, non-BC particles snow RF was the lowest found in our regional analysis. BrC and dust RFs shew a peak occurrence in spring, constituting 13% (+0.014 W/m²) and 12% (+0.011 W/m²) of the total forcing, respectively. RAA forcing was dominated by anthropogenic sources, 94% in winter and 77% in spring, nevertheless, non-BC forcing is likely underestimated. As discussed in Section 3.2, our model underestimated



$f_{non-BC}$ by about a factor of 2 in North America mountain regions, as local dust sources are missed in the model. As a consequence, dust forcing could be larger than estimations reported in our analysis. Doubling BCE attributed to dust in North America, its RF increases by about 80%. Anthropogenic sources contributed for 94% and 77% to RAA forcing in winter and spring, respectively. However, anthropogenic impact should be reduced by increasing dust contribution to the forcing.

In the European continent, total RAA RF was +0.41 and +0.30 W/m$^2$ in winter and spring, respectively. BC absorption represented slightly more than half of the total forcing. BrC forcing was dominated by BF and was located in Eastern regions over Europe (Figure S4). Absorbing OA contributed for about 10% of the total forcing (+0.027 W/m$^2$ in winter and +0.019 W/m$^2$ in spring). The largest non-BC compounds forcing was caused by soil dust. The radiative impact of dust was relevant in Eastern Europe and European Alps snowpack, being the dust snow RF about 30%-40% of the total (+0.027 and +0.019 W/m$^2$ in winter and spring, respectively). Anthropogenic sources explained 67% of European RAA snow RF in winter: their contribution dropped to 58% in spring due to a larger influence of dust.

The largest RAA snow RF values in the middle latitudes was found in Asia. In this area, RAA RF values were +0.56 and +0.64 W/m$^2$ in winter and spring, respectively. BrC contribution was constant (10%, about +0.033 W/m$^2$) between December and May. In particular, BF sources impacted the snow of Northeastern China, Kazakhstan, Southern Russia, and High Mountain, while BB determined some impact in Southern Siberia (Figure S4). Soil dust play a key role in Asia. According to our model, dust snow forcing was +0.10 and +0.24 W/m$^2$ in winter and spring, respectively. Spring dust forcing was larger than BC (+0.19 W/m$^2$) found in the same regions and constituted about a half of the total forcing. As a result, non-BC compounds gave the largest contribution (+60%) to RAA snow RF in Asia during spring. Dust particles caused the anthropogenic sources contribution to all RAA forcing, to drop from 71% in winter, to 41% in springtime. Moreover, it should be noted as Asiatic RAA snow spatio-temporal averaged forcing was low in summer (+0.045 W/m$^2$), with relevant values in the High Mountain region, where it was up to 3 W/m$^2$. About 60% of this forcing was due to non-BC compound, while 10% and 50% is produced by BrC and dust, respectively. Middle latitudes of Asia are the only regions where the snow RAA is relevant in fall. In autumn, total RAA forcing was +0.19 W/m$^2$ and about half of this radiative effect is attributable non-BC compounds (10% and 40% to BrC and dust, respectively).

According to our model, the lowest RAA snow radiative effect has been calculated in Antarctica. Here, the highest values of RF have been found in winter and fall (austral summer and spring), as they were +0.14 and +0.11 W/m$^2$, respectively. The contribution of non-BC compounds was estimated to be 20-30% (10-20% is due to BrC and 10% if from dust), while anthropogenic sources impacted for 68% and 46% in winter and fall, respectively. The role of soil dust in Antarctica snow darkening could be underestimated in this study, as dust emission from arid regions of Southern Hemisphere calculated by our model was 65 Tg/yr. For the same Hemisphere, Ginoux et al. (2012) calculated an emission of 142 Tg/yr, half of which is linked to anthropogenic sources. This mean that dust emission in Austral Hemisphere was underestimated in our simulations by a factor of 2, due to the missing of dust anthropogenic sources. As a result, the presence of anthropogenic dust in the model could increase its role in Antarctica RAA snow RF and reduce the impact of anthropogenic compounds. Moreover, as discussed





in Section 3.1, observations used to evaluate predicted RAA concentration in snow are obsolete and may be not representative of the current black carbon level in Antarctica snow (see Table 3). The most recent measurements of BC, BCE and fnon-BC were carried out in 2012 (Zatko and Warren, 2015) and are relative to sea ice of East Antarctica. Although our modelled RAAs
in snow are in agreement with these observations, this does not imply the same in other regions of Antarctica. As a consequence, it is difficult to establish the level of uncertainty of RF associated to the snow RAA concentration calculation.

### 3.4 Discussion of uncertainties

In this section, the impact of RAA in snow uncertainties on the RF is addressed. The impact of each single uncertainty was evaluated by comparing the perturbed experiments with CTRL simulation. Results are summarized in Table 4.

First, uncertainty of BC, BrC, and dust due to the presence of multiple radiation-absorbing impurities in the snowpack have been assessed. An increasing radiative effect, with respect to CTRL, has been found when RF is calculated for one species at time (OSPT). According to our model, BC snow RF increased by 48% when BrC and dust are not present in the snow. This result is above the upper bound of 10-40%, estimated in previous studies (Flanner et al., 2009; Bond et al., 2013) where only the role of dust was considered in modulating the BC snow RF. BrC forcing was enhanced by 167% when black carbon and
dust are in snowpack. This value is in line with Beres et al. (2020), where a reduction of local BrC RF of about a factor of 2 is found, when the species is added to a dark snowpack. Finally, dust RF increased by 92% in OSPT simulation.

Uncertainty associated to MAC of BC was -18% and +16% when an $E_{abs}$ = 1.1 (BC-L) and $E_{abs}$ = 1.9 (BC-H) are applied to the aged coated particles. The estimated range is comparable with those provided by previous works, where BC forcing uncertainty due to assumptions of its optical properties was estimated in the range ±12% (Bond et al., 2013). BrC forcing
increased by 44% when the blanching of aged BB is missed (BrC-H). Taking into account a whitening process for BF (BrC-L), BrC snow RF is lowered by 22%. The uncertainty of soil dust RF associated to refractive index was -40% and +25% for DUST-L and DUST-H simulations, respectively.

Uncertainties in absorbing optical properties of a given species may also affect the forcing of other RAAs deposited on the snowpack. When BC absorption was increased in BC-H experiment, BrC and dust forcing were reduced by about 10%.
Similarly, in BC-L simulation, the forcing of BrC and dust increased by 10% and 5%, respectively. Uncertainty in BrC absorption properties had a negligible effect (less than 5%) on BC and dust snow radiative effect. By contrast, the perturbation to soil dust absorption optical properties (DUST-H and DUST-L) affected by about 10% BrC snow RF, however, the effect on BC forcing was negligible. The estimated overall uncertainties in RF associated to the absorbing optical properties of RAAs in snow were -18%/+17%, -27%/+45%, -41%/+25%, and -12%/+10% for BC, BrC, soil dust, and total RAAs snow RF.

Uncertainties related to RAA mixing ratio in snow obtained by halving (BCE-L) and doubling (BCE-H) the BCE, were -36%/+50%, -44%/+44%, -35%/+40% for BC, BrC, and mineral dust, respectively. The uncertainty of total RAA snow forcing due to BCE was -36%/+52%.



As for uncertainties linked to $R_e$ (snow ageing), ranges of 30-36% for $R_e$-H and 22-27% for $R_e$-L have been found. RAA snow RF changes due to SCF perturbation were in the range of 22-30% and 11-20% for SCF-H and SCF-L, respectively. The overall

uncertainties in RF due to snowpack properties were -19%/+30%, -25%/+40%, -32%/+30%, and -31%/+46% for BC, BrC, dust and total RAAs.

In summary, the estimated overall uncertainty in RF was -49%/+77% (0.035-0.12 W/m²), -50%/+61% (0.017-0.059 W/m²), -57%/+183% W/m² (0.0028-0.019), -63%/+122% (0.0044-0.025 W/m²), for total RAAs, BC, BrC, and dust, respectively. Total uncertainties were calculated as the root sum of single squared errors, assuming that the single uncertainties are independent.

However, it has to be highlighted as RAA snow RF can be affected by other uncertainties, than what assessed in our study. In particular, the cloud cover, which affect the incident solar radiation at surface and RAA optical properties in snow, also depending on the microlocation of impurities (i.e. if the aerosol particles are externally or internally mixed with snow grains). In the second case, some authors estimated a BC absorption about 1.4-2.1 times larger, with respect to the external mixing (Hansen et al., 2004; Flanner et al., 2012; He et al., 2014). Another uncertainty is related to the snow grain shape: non-spherical

grain assumption reduces the BC snow RF by 20–40% relative to spherical snow grains (He et al., 2014).





## 4 Conclusions

The GEOS-Chem global chemistry and transport model was used to simulate the most relevant radiation-absorbing aerosol
(BC, BrC, and dust) deposition on the snowpack. Moreover, the present-day radiative forcing (RF) due to RAAs in snow has
been calculated. RAA mass concentrations and their optical properties have been simulated, including the most observational
constraints in terms of ageing, size distribution, and absorption properties, following our previous work (Tuccella et al., 2020).

BC and BCE mixing ratios in snow were calculated starting from simulated deposition fields and precipitation fluxes. The
obtained BC and BCE concentrations in snow have been validated through worldwide observations. The model was able to
reproduce the observed regional variations with a $R^2$ of 0.84 and 0.60 for BC and BCE, respectively. 80% of the modelled BC
and BCE values were within a factor of 2 from the observations. The median bias for the same quantities was -13% and -21%.
The model also reproduced the range of observed $f_{non-BC}$ with a $R^2$ of 0.44 and a median bias of -17%.

According to the model, global annual mean present-day RAA snow RF at surface was +0.037, +0.0064, +0.013, and +0.068
W/m² for BC, BrC, dust, and total RAAs, respectively. Non-BC compounds account for 40% (+0.046 W/m²) of RAA snow
global RF and anthropogenic RAAs contribute for 56% (+0.031 W/m²) to the forcing. Total RAA snow RF estimated in this
study is about 6-7 times lower than the direct radiative effect exerted by RAAs (+0.46 W/m²) calculate in our previous study
(Tuccella et al., 2020).

At regional scale, the largest total RAA snow RF was found in Arctic during spring (+0.83 W/m²) and summer (+0.59 W/m²)
and 40% of this forcing was due to non-BC compounds. In particular, non-BC spring RF is mainly due to the dust (+0.12
W/m²), while non-BC was driven by BB BrC (+0.13 W/m²) in summer. In the middle latitudes, the most relevant RAA snow
forcing was obtained in Asia with +0.56 and +0.64 W/m² in winter and spring, respectively. BrC contribution was constant
during winter and spring (10%, about +0.033 W/m²), while soil dust exerted a key role in forcing over Asia: its radiative effect
(+0.24 W/m²) was larger than the one of BC and represented 50% of the total RAA RF in spring. RAA forcing on High
Mountain region was up to 3 W/m² in summertime and 60% of it is attributable to non-BC aerosols. North America exhibited
the lowest RAA snow RF (0.15 and 0.17 W/m² in winter and spring, respectively) and the lowest non-BC contribution (about
20%) in the middle latitudes. As for Europe, total RAA RF was +0.41 and +0.30 W/m² in winter and spring, respectively. BC
contributed slightly more than half of the total forcing. The most relevant non-BC was given by dust (30%-40% of the total).
In Antarctica, the highest values of RF have been found in winter and fall (+0.14 and +0.11 W/m², respectively) and the
contribution of non-BC compounds was estimated to be in the range of 20-30%.

In Arctic, 60% of springtime RF was due to anthropogenic sources, while it dropped down to 30% in summer, due to BB. In
Asia, anthropogenic compounds contributed for the 71% to total forcing in winter, while the contribution was 41% in
springtime, because of the presence of dust. A similar behaviour has been found in Europe, while in North America RAA snow
RF was always dominated by anthropogenic emissions.



Finally, we also explored the sensitivity of RF due to the simultaneous presence of multiple RAAs in snow, absorption optical

properties, uncertainties in impurities mixing ratio, snow grain size and snow coverage. The overall uncertainty in RF associated to these factors were -49%/+77% (0.035-0.12 W/m$^2$), -50%/+61% (0.017-0.059 W/m$^2$), -57%/+183% W/m$^2$ (0.0028-0.019), -63%/+122% (0.0044-0.025 W/m$^2$), for total RAAs, BC, BrC, and dust, respectively.

**Code/Data availability:** GEOS-Chem simulations used in this study are accessible at this link https://osf.io/xntr8/. Snow

impurity measurements are available in the paper referenced in the text. FlexAOD postprocessing tool can be provided upon request to gabriele.curci@aquila.infn.it. RRTMG code is available at this link http://rtweb.aer.com/rrtm_frame.html.

**Author Contributions:** Conceptualization, P.T., G.P.; Methodology, P.T.; Software, P.T., G.C.; Model simulations: P.T.; Formal analysis, P.T., G.P.; writing–original draft preparation, P.T.; writing–review and editing, P.T., V.C., G.P., G.C.; project

administration, V.C., P.T.; funding acquisition, P.T., G.C., and G.P.

**Competing interests**: The authors declare that they have no conflict of interest.

**Funding** Paolo Tuccella is beneficiary of an AXA Research Fund (2016-ENV-PostDoc-University of L'Aquila) postdoctoral

grant. Valentina Colaiuda is supported by PON-AIM (PON-AIM1858058) program funded by Italian Ministry of University and Research (MIUR).





**Table A1.** List of acronyms and symbols.

| | |
|---|---|
| AAE | Absorption Ångström exponent |
| AOD | Aerosol optical depth |
| BB | Biomass burning |
| BC | Black carbon |
| BCE | Black carbon equivalent |
| BF | Biofuel |
| BrC | Brown carbon |
| DEAD | Dust Entrainment And Deposition |
| Eabs | Black carbon absorption enhancement factor |
| FF | Fossil fuel |
| FlexAOD | Flexible aerosol optical depth |
| MAC | Mass absorption coefficient |
| MERRA2 | Modern Era Retrospective-analysis for Research and Application version 2 |
| NIR | Near infrared radiation |
| OA | Organic aerosol |
| POA | Primary organic aerosol |
| $R_e$ | Snow grain effective radius |
| RAA | Radiation-absorbing aerosol |
| RF | Radiative forcing |
| RRTMG | Rapid radiative transfer model for GCM |
| SCF | Snow cover fraction |
| SOA | Secondary organic aerosol |
| VIS | Visible radiation |





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


**Tables**

**Table 1.** Summary of the numerical experiments.

| Experiment | Description |
| --- | --- |
| CTRL[1] | Control run, reference scenario. Mid absorption. |
| OSPT[1] | RF calculated separately for each RAA |
| BC-H[1] | High absorption, aged BC $E_{abs}$=1.9 |
| BC-L[1] | Low absorption, aged BC $E_{abs}$=1.1 |
| BrC-H[1] | High absorption, no blanching for aged BB BrC |
| BrC-L[1] | Low absorption, blanching for aged BF BrC |
| DUST-H[1] | High absorption, dust refractive index from Wagner et al. (2012) |
| DUST-L[1] | Low absorption, dust refractive index from Sinyuk (2003) |
| BCE-H[2] | BCE doubled |
| BCE-L[2] | BCE halved |
| Re-H[2] | Snow grain size multiplied by 2 |
| Re-L[2] | Snow grain size divided by 2 |
| SCF-H[2] | Snow coverage fraction increased by 1.5 |
| SCF-L[2] | Snow coverage fraction decreased by 1.5 |

[1]MACs adopted for these experiments are reported in Table 2.

[2]MACs adopted for these experiments are the same one of CTRL (Table 2).






**Table 2.** Summary of the MACs in visible band[1] used in the experiments. The units are in $m^2/g$.

| Experiment | MAC adopted for each radiation-absorbing aerosol species | | | |
|---|---|---|---|---|
| | Fresh FF BC | Aged FF BC | Fresh BF/BB BC | Aged BF/BB BC |
| CTRL | 6.5 | 9.8 | 6.2 | 9.3 |
| BC-H | 6.5 | 12.4 | 6.2 | 11.8 |
| BC-L | 6.5 | 7.2 | 6.2 | 6.8 |
| | Fresh BF BrC | Aged BF BrC | Fresh BB BrC | Aged BB BrC |
| CTRL | 1.1 | 1.1 | 1.7 | 0.71 |
| BrC-H | 1.1 | 1.1 | 1.7 | 1.7 |
| BrC-L | 1.1 | 0.46 | 1.7 | 0.71 |
| | Dust 0.36–0.6 | Dust 2.6–3.6 | Dust 4.4–6.0 | Dust 7.0–12.0 |
| CTRL | 0.085 | 0.059 | 0.048 | 0.039 |
| DUST-H | 0.14 | 0.086 | 0.067 | 0.052 |
| DUST-L | 0.037 | 0.029 | 0.025 | 0.021 |

[1]MACs reported here are spectrally averaged spectrally averaged between 0.3-0.7 $\mu$m over an incident solar spectrum characteristic of summer high-latitude conditions.

[2]The experiment list is reported in Table 1.




**Table 3.** Comparison of measured and modelled BC and BCE median mixing ratio in surface snow. The units are in ng/g.

| Region | Period | BC Observed | BC Modelled | BCE Observed | BCE Modelled |
|---|---|---|---|---|---|
| *Arctic[1]* | | | | | |
| Arctic Ocean, spring | 2005-2008 | 7±3 | 9 | 12±5 | 16 |
| Arctic Ocean, summer | 2005-2008 | 8±8 | 7 | 14±15 | 11 |
| Canadian and Alaskan Arctic | Apr.-May 2007-2009 | 8±3 | 7 | 14±7 | 11 |
| Canadian sub-Arctic | Mar.-Apr. 2009 | 14±9 | 8 | 20±12 | 12 |
| Greenland, spring | Apr. 2009 | 4±2 | 6 | 7±3 | 8 |
| Greenland, summer | 2006-2008 | 1±1 | 7 | 3±3 | 11 |
| Western Russia | Mar.-May 2007 | 27* (12-48)** | 18 | 34* (15-60)** | 32 |
| Eastern Russia | Mar.-May 2008 | 34±46 | 11 | 48±90 | 20 |
| Svalbard | Mar.-Apr. 2007, 2009 | 13±9 | 11 | 18±12 | 15 |
| Tromso, Norway | May 2008 | 21±12 | 17 | 29±16 | 25 |
| *Antarctica* | | | | | |
| Vostok[2] | Dec. 1990-Feb. 1991 | 0.60* | 0.61 | - | - |
| Simple Dome[3] | 1982-1985 | 2.5 (2.3-2.9)** | 0.38 | - | - |
| South Pole[4] | Jan.-Feb. 1996 | 0.23* (0.10-0.34)** | 0.37 | - | - |
| Sea Ice[5] | Sep.-Nov. 2012 | 0.30±0.20 | 0.53 | 0.40±0.30 | 0.80 |
| *North America[6]* | | | | | |
| Pacific Northwest | Jan.-Mar. 2013 | 22±44 | 13 | 29±52 | 15 |
| Intramountain Northwest | Jan.-Mar. 2013 | 24±34 | 28 | 37±93 | 35 |
| North U.S. Plains | Jan.-Mar. 2013 | 30±54 | 37 | 78±245 | 39 |
| Canada | Jan.-Mar. 2013 | 15±13 | 15 | 25±45 | 18 |
| *Northwest China[7]* | | | | | |
| Northern Xinjiang | Jan.-Feb. 2012 | 73 ± 120 | 61 | - | - |
| *Northeast China[8]* | | | | | |
| Qilian Mountains | Jan.-Feb. 2010 | - | - | 1550* (426-3042)** | 493 |
| Inner Mongolia | Jan.-Feb. 2010 | 340±910 | 338 | 820±3060 | 1057 |
| Northeast border | Jan.-Feb. 2010 | 135* (68-295)** | 68 | 190* (100-374)** | 98 |
| Industrial Northeast | Jan.-Feb. 2010 | 1220±600 | 436 | 1720±840 | 556 |
| *Himalayas and Tibet Plateau[9]* | | | | | |
| Hilamalayas, summer | 2000-2001 | 21* (0.3-43)** | 48 | - | - |
| Tibet Plateau, summer | 2001 | 45* (18-446)** | 26 | - | - |

[1]Doherty et al. (2010)

[2]Grenfell et al. (1994)

[3]Chylek et al. (1987)





[4]Warren and Clark (1990)

[5]Zatko and Warren (2015)

[6]Doherty et al. (2014)

[7]Ye et al. (2012)

[8]Wang et al. (2013)

[9]Kopacz et al. (2011)

[*]Average

[**] The standard deviations are not available. The values in the brackets represent the low-high range measured in the region.

**Table 4.** Global all-sky annual mean surface snow RF (W/m$^2$) of total RAAs, BC, BrC, and mineral dust calculated in the
experiments discussed in Section 2.4 (see also Table 1). The percentages represent the deviations from CTRL run.

| Experiment | All RAAs | BC | BrC | Dust |
|---|---|---|---|---|
| CTRL | +0.068 | +0.033 | +0.0066 | +0.012 |
| OSPT[1] | +0.089 (+31%) | +0.049 (+48%) | +0.018 (+167%) | +0.023 (+92%) |
| BC-H | +0.073 (+8%) | +0.038 (+16%) | +0.0059 (-11%) | +0.011 (-10%) |
| BC-L | +0.062 (-9%) | +0.027 (-18%) | +0.0073 (+10%) | +0.013 (+5%) |
| BrC-H | +0.071 (+4%) | +0.032 (-2%) | +0.0095 (+44%) | +0.012 (-) |
| BrC-L | +0.067 (-2%) | +0.034(+4%) | +0.0051 (-22%) | +0.012 (-) |
| DUST-H | +0.071 (+5%) | +0.032 (-4%) | +0.0059 (-11%) | +0.015 (+25%) |
| DUST-L | +0.063 (-7%) | +0.035 (+5%) | +0.0071 (+8%) | +0.0072 (-40%) |
| BCE-H | +0.10 (+52%) | +0.050 (+50%) | +0.0095 (+44%) | +0.017 (+40%) |
| BCE-L | +0.044 (-36%) | +0.021 (-36%) | +0.0037 (-44%) | +0.0078 (-35%) |
| $R_e$-H | +0.092 (+36% ) | +0.045 (+36%) | +0.0088 (+33%) | +0.016 (+30%) |
| $R_e$-L | +0.050 (-27%) | +0.024 (-27%) | +0.0051 (-22%) | +0.0090 (-25%) |
| SCF-H | +0.088 (+29%) | +0.043 (+29%) | +0.0081 (+22%) | +0.016 (+30%) |
| SCF-L | +0.058 (-15%) | +0.029 (-13%) | +0.0059 (-11%) | +0.0096 (-20%) |
| Total Uncertain[2] | 0.035- 0.12 | 0.017-0.059 | 0.0028-0.019 | 0.0044-0.025 |
| | -49%/+77% | -50%/+61% | -57%/+183% | -63%/+112% |

[1]For this experiment, total RAAs snow RF was calculated as the sum of the single species.

[2] The lower and upper bounds were calculated by adding in quadrature the RF from each experiment.



**Table 5. Domain of the regions used in this study.**

| Region | Longitude range | Latitude range |
| --- | --- | --- |
| Arctic | 60°N - 90 °N | -180°E – 180°E |
| Nord America | 29°N – 60°N | -155°E – 60°E |
| Europe | 40°N – 60°N | -10°E – 45°E |
| Asia | 25°N – 60°N | 45°E – 160°E |
| Antarctica | 90°S – 60°S | -180°E – 180°E |






**Figures**

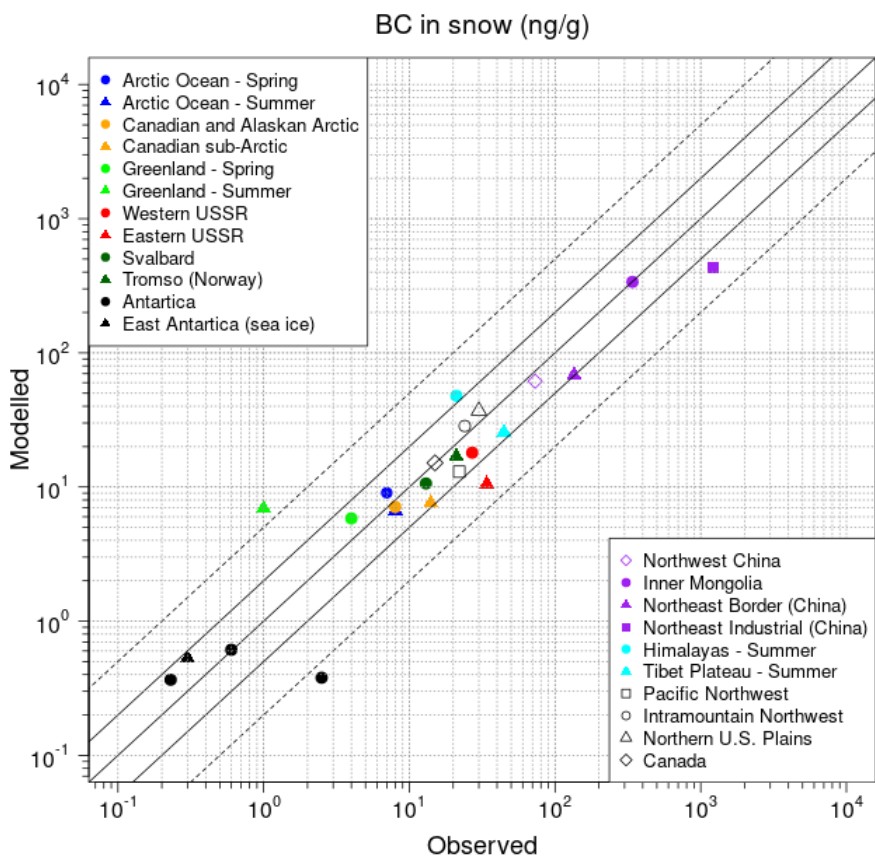

**Figure 1: Scatter plots of the observed and modelled BC mixing ratio in the snow. Central continue line is the 1:1 line, other continue lines correspond to 1:2 and 2:1 lines. Dotted lines correspond to 1:5 and 5:1 lines.**





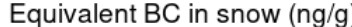

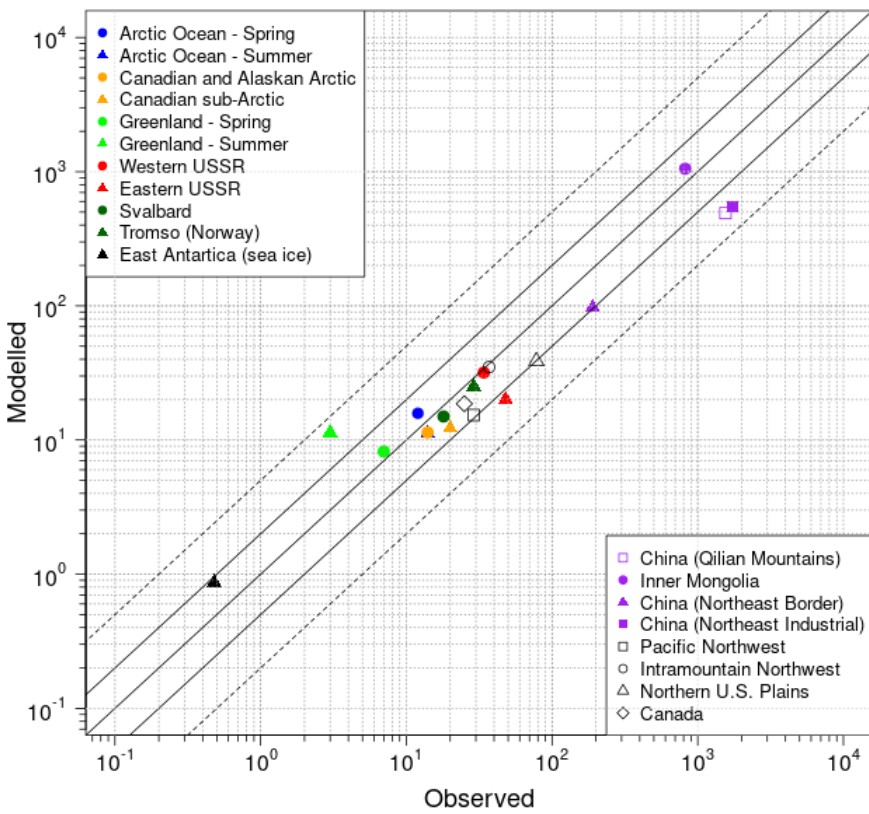

**Figure 2. Same as Figure 1, but for black carbon equivalent (BCE). BCE is defined in Equation 2 and represents the RAA snow mass scaled with the MAC of each species.**



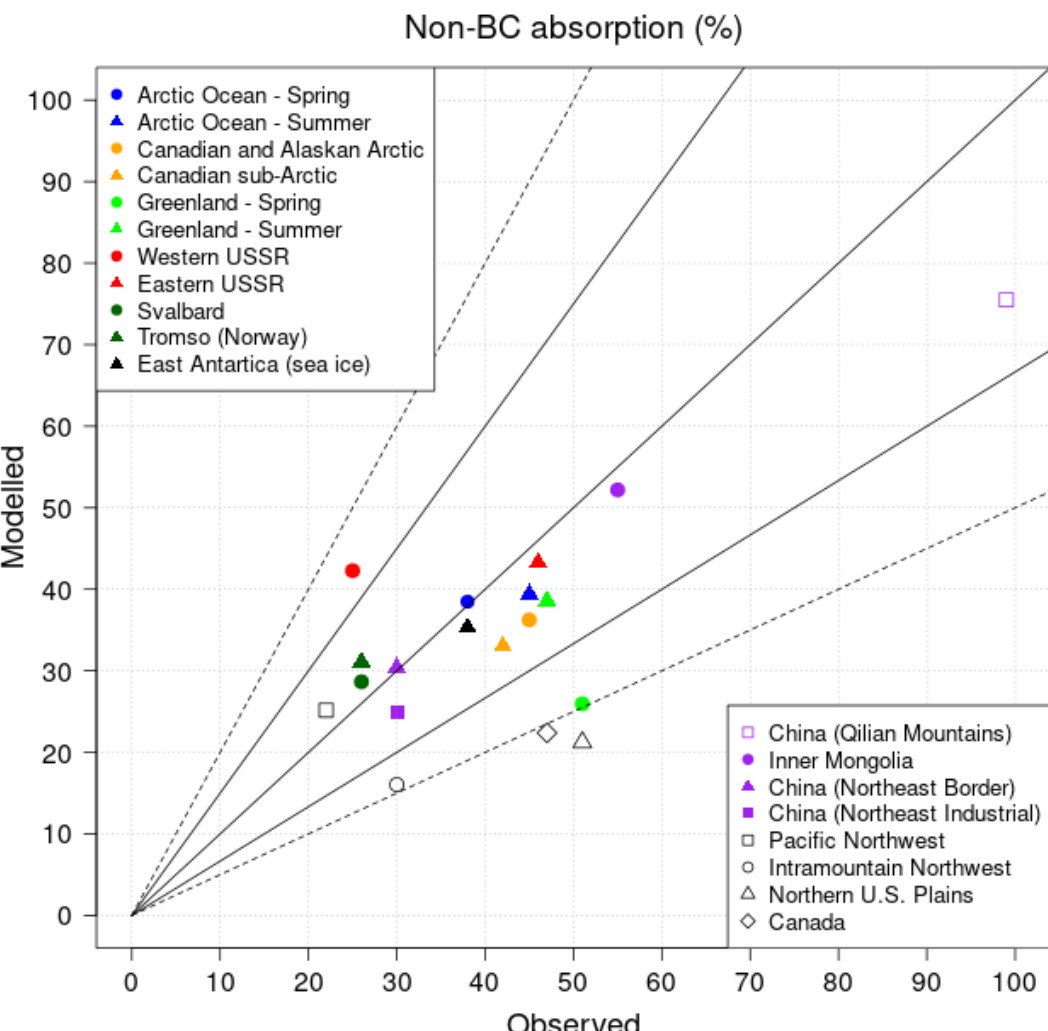

**Figure 3: Scatter plots of the observed and modelled $f_{\text{non-BC}}$ in the snow. Central continue line is the 1:1 line, other continue lines correspond to 1:1.5 and 1.5:1 lines. Dotted lines correspond to 1:2 and 2:1 lines. Non-BC means the sum of BrC and dust (reported to BC equivalent).**





**Figure 4. All-sky annual mean (2010–2014) radiation-absorbing aerosols (RAA), black carbon (BC), brown carbon (BrC), and soil dust in snow radiative forcing (RF) calculated from CTRL experiment.**

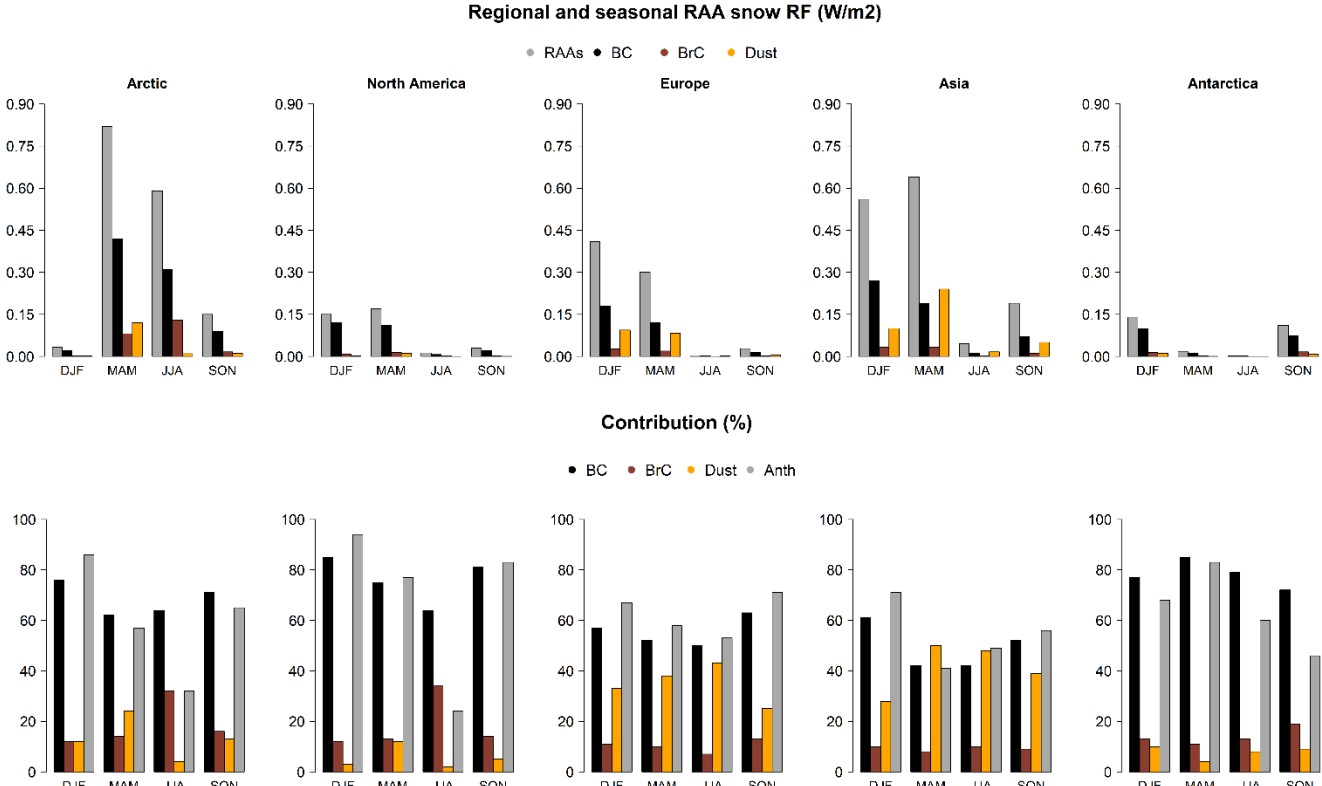

**Figure 5. Top panel: all-sky regional and seasonal averages (2010–2014) (top panel) of total RAAs, black carbon (BC), brown carbon (BrC), and soil dust snow radiative forcing (RF), calculated from CTRL experiment in Arctic, North America, Europa, Asia, and Antarctica. Lower panel: contribution of each single species and anthropogenic RAA to total forcing. The anthropogenic contribution is given by BC and BrC from fossil fuel (FF) and biofuel (BF) sources and aromatic SOA.**