# Peer review of "Present-day radiative effect from radiation-absorbing aerosols in snow"

_Atmospheric Chemistry and Physics, 2020_

## Referee Comment (RC1) · Anonymous Referee #3 · 21 Nov 2020

Recommendation: Major revision

General comments

By performing a set of simulations using the GEOS-Chem global chemistry and transport model this study investigates the present-day radiative forcing (RF) of radiation absorbing aerosols (RAA), namely Black carbon (BC), brown carbon (BrC) and soil dust, in snow. The study quantifies global mean RF for different RAA species and its seasonality. The study further analyses some regional characteristics of RF for different RAA species and estimated it uncertainty. The topic is an interesting one and the paper is worth of publication. However, there are some specific comments listed below that need to be addressed to improve the quality of the paper. The paper, therefore, needs a major revision before it can be acceptable for publication.

[Figure]

Major comments

1. What are new results or advances of understanding RF of RAA in this study in comparison with existing literatures? This aspect shall be improved in abstract and conclusion. 2. What are implications of this study? Some comments on this would improve the quality of the paper. 3. Section 2 on Method is too long (from page 4 to page 10) and there is no need for such detailed descriptions in each sub-section in this part. This part needs to be shortened considerably. 4. Authors need to read the paper carefully and to check English usages and possible errors.

Minor comments

1. Lines 9-10 on page 1. "a 5-years simulation". Shall be "a set of 5-year simulations"? 2. Line 296. "these information" to "this information" 3. Line 299. "we the 5-year" to "the 5-year". 4. Line 368. "shew". Do you mean "showed"? Please also check other places. 5. Font used in panels of Figure S3 is too small. Suggest increasing font size or giving more detailed figure caption.
* * *

---

## Referee Comment (RC2) · Anonymous Referee #4 · 31 Jan 2021

**Review of article: "Present-day radiative effect from radiation-absorbing aerosols in snow"** *by Paolo Tuccella, Giovanni Pitari, Valentina Colaiuda, Gabriele Curci*

This work presents an estimation of present-day radiative forcing (RF) of radiation absorbing aerosols (Black carbon-BC, brown carbon-BrC and soil dust) by using the GEOS-Chem global chemistry and transport model. First, the authors performed an evaluation of their simulations. Then, they estimated a global mean all sky RF of 0.068, 0.033, 0.0066, and 0.012 W/m2 for total RAA, BC, BrC and dust snow. Non-BC is the compound with the highest contribution to snow RF and anthropogenic compounds account for 56% of the RF. Moreover, seasonal and spatial differences regarding the specie are described. Authors also estimated the uncertainties in snow RF due to mixing ratio in snow, snow grain dimension and snow cover fraction.

In my opinion the paper presents an interesting work and the quality for the publication in this review. However, I have found some issues that deserve a minor revision or technical correction and could, in my opinion, improve the overall quality of this work.

My detailed comments are given below.

Methods
- Lines 102-107. Please include the references for the EDGAR v4.2 database, the RETRO inventory and the different regional inventories used.
- Line 197: "We have used Two different […]" correct by "We have used two different […]".
- Line 236: "[…], for this reasontwo extreme […]" correct by "[…], for this reason two extreme […]"

Results
In my opinion, this section would benefit from a change in the name by "Results and discussion" because both the presentation and the discussion of the results is performed here.
- In my opinion, Figure 1 and 2 are enough to show the evaluation results, table 3 may move to supplementary material.
- Please correct "Angstrom" by "Ångström".
- Lines 333-334: "In this case, the analysis of the light fraction absorption due to non-BC compounds (fnon-BC) revealed other aspects of the model skill in reproducing the RAA in snow." What are these aspects? Please clarify.
- Line 335: "In Figure 3 a comparison between calculated and observed fnon-BC. fnon-BC is shown proposed," I think there is some mistake in this statement.
- Lines 358-359: "In Table 4 a summary of the RF estimated from all numerical experiments discussed in Section 2.4 is given." This should be indicated in section 3.4 as is done and not here. Please, remove.

- Line 368 and 411: "BF BC shew an impact […]" and "BrC and dust RFs shew a peak" would you mean "show" instead of shew?
- Caption figure 5: "[…]. Lower panel: […]" correct by "[…]. Bottom panel: […]"

Conclusions

A paragraph discussion the novelty of this work and how this work will help to the actual knowledge would improve the quality of the work.

---

## Author Comment (AC1) · 17 Mar 2021

Answer to Reviewer #1 for the article submitted to "Atmospheric Chemistry and Physics":

**"Present-day radiative effect from radiation-absorbing aerosols in snow"**

by Paolo Tuccella, Giovanni Pitari, Valentina Colaiuda, Edoardo Raparelli, Gabriele Curci

Dear Reviewer and Editor,

We acknowledge the Reviewer for the time spent to evaluate our work and we thank him/her for the useful and constructive comments. We also acknowledge the Editor and we made all proposed changes in the revised manuscript. Note that our answers are in blue in the following text. Please also note that we have included a new author (Edoardo Raparelli) in the revised paper.

By performing a set of simulations using the GEOS-Chem global chemistry and transport model this study investigates the present-day radiative forcing (RF) of radiation absorbing aerosols (RAA), namely Black carbon (BC), brown carbon (BrC) and soil dust, in snow. The study quantifies global mean RF for different RAA species and its seasonality. The study further analyses some regional characteristics of RF for different RAA species and estimated it uncertainty. The topic is an interesting one and the paper is worth of publication. However, there are some specific comments listed below that need to be addressed to improve the quality of the paper. The paper, therefore, needs a major revision before it can be acceptable for publication.

Major comments

1. What are new results or advances of understanding RF of RAA in this study in comparison with existing literatures? This aspect shall be improved in abstract and conclusion. 2. What are implications of this study? Some comments on this would improve the quality of the paper.

Many thanks for these suggestions. In order to clarify the points raised by Reviewer, we have some paragraphs in the text:

- At the end of introduction, we have added a paragraph the content of each section:

  *"RAAs mass and their optical properties have been simulated using the most recent updates in terms of ageing, size distribution and absorption optical properties, inferred from observational constraints following our previous work (Tuccella et al., 2020). Starting from the GEOS-Chem output, we have diagnosed the mass mixing ratios of RAAs in snow and, subsequently, calculated the RF for different RAA species. In Section 2, we provided the description of modelling tools used in this study. In Section 3, the modelled RAA content in snow was compared with the available observations and the associated RF was, therefore, calculated, taking into account the simultaneous presence of BC, BrC, and dust in the snow. Moreover, seasonal and regional differences about the RF of each species have been explored. We also provided insights about the contribution of anthropogenic and carbonaceous compounds to the total RF. Finally, we discussed the uncertainties in modelling this kind of forcing associated with the assumptions of RAA optical properties, RAA mixing ratio uncertainty, snow ageing, and snow cover fraction. The conclusions are given in Section 4."*.

- At the end of subsection 3.2, we have added the following paragraph which help us to support the implication of this study:

  *"Total RAA-snow RF estimated in this study is about 6-7 times lower than the direct radiative effect exerted by RAAs (+0.36 and +0.10 $W/m^2$ for BC+BrC mixture and dust, respectively), calculated in our previous study (Tuccella et al., 2020). However, comparing RAA forcings in atmosphere and in snow by scaling them with their efficacies (Hansen et al., 2005), they are of the same order of magnitude. Forcing efficacy for BC in snow has been estimated 3 times larger than the one resulting from $CO_2$ (Flanner et al., 2007; Bond et al., 2013; Boucher et al., 2013). Assuming that the same*

*efficacy is valid for both BrC and dust, the effective present-day RF from RAA in snow obtained in this study is +0.20 W/m². The atmospheric forcing estimated in Tuccella et al. (2020) may be scaled with the efficacies reported by Hansen et al. (2005), resulting in a total RAA effective forcing near +0.30 W/m²."*

- In subsection 3.3, after having calculating the overall uncertainty in RF, we have added the following sentences which support the implication of this study:

*"Our results indicate that the lower bounds of total uncertainty of BC, BrC and dust were comparable. By contrast, upper bounds for BrC and dust were about 2 and 3 times larger than the one of BC. According to our calculation, this uncertainty was related to the simultaneous presence of multiple RAA species in the snowpack (OSPT experiment)."*

- In the conclusions, we have changed the first as follows:

*"We presented a global modelling study to assess the present-day RF in snow due to the most relevant radiation-absorbing aerosols (BC, BrC, and dust). While BC RF in snow has been extensively studied (e.g., Bond et al. 2013; Boucher et al., 2013), the forcing from BrC and dust and associated uncertainties were not assessed in IPCC AR5, although they are recognised as radiation-absorbing particles. As a consequence, it is not clear what is the contribution of anthropogenic sources and carbonaceous aerosols to RAA-snow RF. Moreover, given that the snow albedo change is not linear with the impurity content (Flanner et al., 2009; Dang et al., 2017), RF from RAAs in snow has to be calculated taking simultaneously into account the concentrations of BC, BrC, and mineral soil dust."*

We have also changed the last part as:

*"Finally, we also explored the sensitivity of RF due to the simultaneous presence of multiple RAAs in snow, absorption optical properties, uncertainties in impurities mixing ratio, snow grain size and snow coverage. The overall uncertainty in RF associated to these factors were -49%/+77% (0.035-0.12 W/m²), -50%/+61% (0.017-0.059 W/m²), -57%/+183% W/m² (0.0028-0.019), -63%/+122% (0.0044-0.025 W/m²), for total RAAs, BC, BrC, and dust, respectively. **These results highlight that uncertainty upper bounds of BrC and dust were about 2 and 3 times larger than the one of BC. This uncertainty was mainly due to the simultaneous presence of multiple absorbing impurities in the snow. Therefore, we may conclude that RAA snow RF is very sensitive to the concomitant presence of more species, especially for non-BC compounds, given their minor absorption with respect to BC.***

***Efficacy of RF associated to BC in snow was 3 times larger than forcing from CO₂ (Flanner et al., 2007; Bond et al., 2013; Boucher et al., 2013). Assuming the same efficacy for BrC and dust, effective RF exerted by RAA in snow found in this study was +0.20 W/m², a value comparable with the RAA effective atmospheric forcing (about +0.30 W/m²) obtained from Tuccella et al. (2020). Given that RF of RAAs in snow acts mainly on the cryosphere, it may potentially have important effects in response to the snow-albedo feedback. As a consequence, a reduction of the uncertainties is desirable. According to our results, a first step to reduce uncertainties in RAA-snow RF should be an improvement of the representation of RAAs in snow within models, through constraint with local and satellite observations and a better characterization of the emission inventories in current atmospheric models."***

- Finally, we have changed the abstract as follows adding some sentences (in bold):

*"Black carbon (BC), brown carbon (BrC) and soil dust are the most important radiation absorbing aerosols (RAA). When RAA are deposited on the snowpack, they lower the snow albedo causing an*

*increase of the solar radiation absorption. The climatic impact associated to the snow darkening induced by RAA is highly uncertain.* **The climatic impact associated to the deposition of BrC and dust on snowpack and its uncertainties were not reported in the IPCC 5th Assessment Report (AR5), therefore, the contribution of anthropogenic sources and carbonaceous aerosols to RAA radiative forcing (RF) in snow is not clear. Moreover, the snow albedo perturbation induced to a single RAA species depends on the presence of other light-absorbing impurities contained in the snowpack. In this work, we calculated the present-day RF of RAA in snow starting from the deposition fields from a 5-years simulation with the GEOS-Chem global chemistry and transport model.** *RF was estimated taking into account the presence of BC, BrC, and mineral soil dust in snow, simultaneously. Modelled BC and black carbon equivalent (BCE) mixing ratios in snow and the fraction of light absorption due to non-BC compounds ($f_{non-BC}$) were compared with worldwide observations. We showed as BC, BCE and $f_{non-BC}$, obtained from deposition and precipitation fluxes, reproduce the regional variability and order of magnitude of the observations. Global average all-sky total RAA, BC, BrC and dust snow RF were 0.068, 0.033, 0.0066, and 0.012 W/m$^2$, respectively. At global scale, non-BC compounds accounted for 40% of RAA snow RF, while anthropogenic RAAs contributed to the forcing for 56%. With regard to non-BC compounds, the largest impact of BrC has been found during summer in the Arctic (+0.13 W/m$^2$). In the middle latitudes of Asia, the forcing from dust in spring accounted for the 50% (+0.24 W/m$^2$) of the total RAAs RF. Uncertainties in absorbing optical properties, RAA mixing ratio in snow, snow grain dimension, and snow cover fraction resulted in an overall uncertainty of -50%/+61%, -57%/+183%, -63%/+112%, and -49%/+77% in BC, BrC, dust and total RAAs snow RF, respectively.* **Uncertainty upper bounds of BrC and dust were about 2 and 3 times larger than the upper bounds associated to BC. Higher BrC and dust uncertainties were mainly due to the presence of multiple absorbing impurities in the snow. Our results highlight that an improvement of the representation of RAAs in snow is desirable, given the potential high efficacy of this forcing.**

3. Section 2 on Method is too long (from page 4 to page 10) and there is no need for such detailed descriptions in each sub-section in this part. This part needs to be shortened considerably.

We have reduced GEOS-Chem description, the section describing RAA optical properties, and the section about the radiative transfer model. The details have been moved in the Supplement. Further, we have removed some paragraphs and sentences from sections about snow albedo perturbation and numerical experiments. We think that the descriptions provided in these sections in the original draft are useful to justify our sensitivity simulations.

4. Authors need to read the paper carefully and to check English usages and possible errors.

English has been carefully revised as suggested.

Minor comments

1. Lines 9-10 on page 1. "a 5-years simulation". Shall be "a set of 5-year simulations"?

Herein, we were referring to the GEOS-Chem simulation. This run is a unique 5-year long and it was aimed to simulate the aerosol concentration in atmosphere and deposition field. Starting from this output, we carried out the set of numerical experiments described in the text. We have rephrased the sentence as follows:

*"In this work, the present-day radiative forcing (RF) of RAA in snow was calculated starting from deposition fields calculated with a 5-years simulation with GEOS-Chem global chemistry and transport model."*

2. Line 296. "these information" to "this information"

The sentence has been corrected as suggested.

3. Line 299. "we the 5-year" to "the 5-year".

The sentence has been corrected as suggested.

4. Line 368. "shew". Do you mean "showed"? Please also check other places.

This has been corrected through the text.

5. Font used in panels of Figure S3 is too small. Suggest increasing font size or giving more detailed figure caption

Figure S3 has been divided in 3 new figures S3, S4, and, S5, for BrC-BF, BrC-BB, and BrC-SOA, respectively.

---

## Author Comment (AC2) · 17 Mar 2021

Answer to Reviewer #2 for the article submitted to "Atmospheric Chemistry and Physics":

**"Present-day radiative effect from radiation-absorbing aerosols in snow"**

by Paolo Tuccella, Giovanni Pitari, Valentina Colaiuda, Edoardo Raparelli, Gabriele Curci

Dear Reviewer and Editor,

We acknowledge the Reviewer for the time spent to evaluate our work and we thank him/her for the useful and constructive comments. We also acknowledge the Editor and we made all proposed changes in the revised manuscript. Note that our answers are in blue in the following text. Please also note that we have included a new author (Edoardo Raparelli) in the revised paper.

This work presents an estimation of present-day radiative forcing (RF) of radiation absorbing aerosols (Black carbon-BC, brown carbon-BrC and soil dust) by using the GEOS-Chem global chemistry and transport model. First, the authors performed an evaluation of their simulations. Then, they estimated a global mean all sky RF of 0.068, 0.033, 0.0066, and 0.012 W/m$^2$ for total RAA, BC, BrC and dust snow. Non-BC is the compound with the highest contribution to snow RF and anthropogenic compounds account for 56% of the RF. Moreover, seasonal and spatial differences regarding the species are described. Authors also estimated the uncertainties in snow RF due to mixing ratio in snow, snow grain dimension and snow cover fraction.

In my opinion the paper presents an interesting work and the quality for the publication in this review. However, I have found some issues that deserve a minor revision or technical correction and could, in my opinion, improve the overall quality of this work.

My detailed comments are given below.

Methods
- Lines 102-107. Please include the references for the EDGAR v4.2 database, the RETRO inventory and the different regional inventories used.
  Following the recommendation of Reviewer 1, in order to shorten the section 2, we have moved the GEOS-Chem description in the Supplement. The references for emission inventories have been included as suggested.

- Line 197: "We have used Two different […]" correct by "We have used two different […]".
  The sentence has been corrected as suggested.

- Line 236: "[…], for this reasontwo extreme […]" correct by "[…], for this reason two extreme […]"
  The sentence has been corrected as suggested.

Results
- In my opinion, this section would benefit from a change in the name by "Results and discussion" because both the presentation and the discussion of the results is performed here.
  The suggestion to rename this section a "Results and discussion" is welcome.

- In my opinion, Figure 1 and 2 are enough to show the evaluation results, table 3 may move to supplementary material.
  Table 3 has been moved in the Supplement as recommended.

- Please correct "Angstrom" by "Ångström".
  Angstrom has been corrected as Ångström.

- Lines 333-334: "In this case, the analysis of the light fraction absorption due to non-BC compounds (fnon-BC) revealed other aspects of the model skill in reproducing the RAA in snow." What are these aspects? Please clarify.

Here, we mean other aspect in terms of biases related to the emissions, transport, and assumptions done for absorbing optical properties. The sentence has been rephrased as follows:

*"In this case, the analysis of the light fraction absorption due to non-BC compounds (fnon-BC) revealed other aspects of the model skill in reproducing the RAA in snow, in terms of biases related to the emissions, transport, and assumptions done for absorbing optical properties."*

- Line 335: "In Figure 3 a comparison between calculated and observed fnon- BC. fnon-BC is shown proposed," I think there is some mistake in this statement.
  Yes, there is a mistake. We have corrected the sentence.

- Lines 358-359: "In Table 4 a summary of the RF estimated from all numerical experiments discussed in Section 2.4 is given." This should be indicated in section 3.4 as is done and not here. Please, remove.
  The sentence has been removed as suggested.

- Line 368 and 411: "BF BC shew an impact […]" and "BrC and dust RFs shew a peak" would you mean "show" instead of shew?
  "Shew" has been changed in "showed".

- Caption figure 5: "[…]. Lower panel: […]" correct by "[…]. Bottom panel: […]"
  The caption has been corrected as suggested.

Conclusions

A paragraph discussion the novelty of this work and how this work will help to the actual knowledge would improve the quality of the work.

Many thanks for this recommendation. Also following the suggestion of Reviewer 1, this aspect has been improved in several part of the text.

- At the end of introduction, we have added a paragraph the content of each section:

  *"RAAs mass and their optical properties have been simulated using the most recent updates in terms of ageing, size distribution and absorption optical properties, inferred from observational constraints following our previous work (Tuccella et al., 2020). Starting from the GEOS-Chem output, we have diagnosed the mass mixing ratios of RAAs in snow and, subsequently, calculated the RF for different RAA species. In Section 2, we provided the description of modelling tools used in this study. In Section 3, the modelled RAA content in snow was compared with the available observations and the associated RF was, therefore, calculated, taking into account the simultaneous presence of BC, BrC, and dust in the snow. Moreover, seasonal and regional differences about the RF of each species have been explored. We also provided insights about the contribution of anthropogenic and carbonaceous compounds to the total RF. Finally, we discussed the uncertainties in modelling this kind of forcing associated with the assumptions of RAA optical properties, RAA mixing ratio uncertainty, snow ageing, and snow cover fraction. The conclusions are given in Section 4.".*

- At the end of subsection 3.2, we have added the following paragraph which help us to support the implication of this study:

  *"Total RAA-snow RF estimated in this study is about 6-7 times lower than the direct radiative effect exerted by RAAs (+0.36 and +0.10 $W/m^2$ for BC+BrC mixture and dust, respectively), calculated in our previous study (Tuccella et al., 2020). However, comparing RAA forcings in atmosphere and in snow by scaling them with their efficacies (Hansen et al., 2005), they are of the same order of magnitude. Forcing efficacy for BC in snow has been estimated 3 times larger than the one resulting from $CO_2$ (Flanner et al., 2007; Bond et al., 2013; Boucher et al., 2013). Assuming that the same*

*efficacy is valid for both BrC and dust, the effective present-day RF from RAA in snow obtained in this study is +0.20 W/m². The atmospheric forcing estimated in Tuccella et al. (2020) may be scaled with the efficacies reported by Hansen et al. (2005), resulting in a total RAA effective forcing near +0.30 W/m²."*

- In subsection 3.3, after having calculating the overall uncertainty in RF, we have added the following sentences which support the implication of this study:

*"Our results indicate that the lower bounds of total uncertainty of BC, BrC and dust were comparable. By contrast, upper bounds for BrC and dust were about 2 and 3 times larger than the one of BC. According to our calculation, this uncertainty was related to the simultaneous presence of multiple RAA species in the snowpack (OSPT experiment)."*

- In the conclusions, we have changed the first as follows:

*"We presented a global modelling study to assess the present-day RF in snow due to the most relevant radiation-absorbing aerosols (BC, BrC, and dust). While BC RF in snow has been extensively studied (e.g., Bond et al. 2013; Boucher et al., 2013), the forcing from BrC and dust and associated uncertainties were not assessed in IPCC AR5, although they are recognised as radiation-absorbing particles. As a consequence, it is not clear what is the contribution of anthropogenic sources and carbonaceous aerosols to RAA-snow RF. Moreover, given that the snow albedo change is not linear with the impurity content (Flanner et al., 2009; Dang et al., 2017), RF from RAAs in snow has to be calculated taking simultaneously into account the concentrations of BC, BrC, and mineral soil dust."*

We have also changed the last part as:

*"Finally, we also explored the sensitivity of RF due to the simultaneous presence of multiple RAAs in snow, absorption optical properties, uncertainties in impurities mixing ratio, snow grain size and snow coverage. The overall uncertainty in RF associated to these factors were -49%/+77% (0.035-0.12 W/m²), -50%/+61% (0.017-0.059 W/m²), -57%/+183% W/m² (0.0028-0.019), -63%/+122% (0.0044-0.025 W/m²), for total RAAs, BC, BrC, and dust, respectively.* **These results highlight that uncertainty upper bounds of BrC and dust were about 2 and 3 times larger than the one of BC. This uncertainty was mainly due to the simultaneous presence of multiple absorbing impurities in the snow. Therefore, we may conclude that RAA snow RF is very sensitive to the concomitant presence of more species, especially for non-BC compounds, given their minor absorption with respect to BC.**

**Efficacy of RF associated to BC in snow was 3 times larger than forcing from $CO_2$ (Flanner et al., 2007; Bond et al., 2013; Boucher et al., 2013). Assuming the same efficacy for BrC and dust, effective RF exerted by RAA in snow found in this study was +0.20 W/m², a value comparable with the RAA effective atmospheric forcing (about +0.30 W/m²) obtained from Tuccella et al. (2020). Given that RF of RAAs in snow acts mainly on the cryosphere, it may potentially have important effects in response to the snow-albedo feedback. As a consequence, a reduction of the uncertainties is desirable. According to our results, a first step to reduce uncertainties in RAA-snow RF should be an improvement of the representation of RAAs in snow within models, through constraint with local and satellite observations and a better characterization of the emission inventories in current atmospheric models."**

- Finally, we have changed the abstract as follows adding some sentences (in bold):

*"Black carbon (BC), brown carbon (BrC) and soil dust are the most important radiation absorbing aerosols (RAA). When RAA are deposited on the snowpack, they lower the snow albedo causing an*

*increase of the solar radiation absorption. The climatic impact associated to the snow darkening induced by RAA is highly uncertain.* **The climatic impact associated to the deposition of BrC and dust on snowpack and its uncertainties were not reported in the IPCC 5th Assessment Report (AR5), therefore, the contribution of anthropogenic sources and carbonaceous aerosols to RAA radiative forcing (RF) in snow is not clear. Moreover, the snow albedo perturbation induced to a single RAA species depends on the presence of other light-absorbing impurities contained in the snowpack. In this work, we calculated the present-day RF of RAA in snow starting from the deposition fields from a 5-years simulation with the GEOS-Chem global chemistry and transport model.** *RF was estimated taking into account the presence of BC, BrC, and mineral soil dust in snow, simultaneously. Modelled BC and black carbon equivalent (BCE) mixing ratios in snow and the fraction of light absorption due to non-BC compounds ($f_{non-BC}$) were compared with worldwide observations. We showed as BC, BCE and $f_{non-BC}$, obtained from deposition and precipitation fluxes, reproduce the regional variability and order of magnitude of the observations. Global average all-sky total RAA, BC, BrC and dust snow RF were 0.068, 0.033, 0.0066, and 0.012 $W/m^2$, respectively. At global scale, non-BC compounds accounted for 40% of RAA snow RF, while anthropogenic RAAs contributed to the forcing for 56%. With regard to non-BC compounds, the largest impact of BrC has been found during summer in the Arctic (+0.13 $W/m^2$). In the middle latitudes of Asia, the forcing from dust in spring accounted for the 50% (+0.24 $W/m^2$) of the total RAAs RF. Uncertainties in absorbing optical properties, RAA mixing ratio in snow, snow grain dimension, and snow cover fraction resulted in an overall uncertainty of -50%/+61%, -57%/+183%, -63%/+112%, and -49%/+77% in BC, BrC, dust and total RAAs snow RF, respectively.* **Uncertainty upper bounds of BrC and dust were about 2 and 3 times larger than the upper bounds associated to BC. Higher BrC and dust uncertainties were mainly due to the presence of multiple absorbing impurities in the snow. Our results highlight that an improvement of the representation of RAAs in snow is desirable, given the potential high efficacy of this forcing.**

---

## Author Response (AR1)

Dear Reviewers, Dear Editor,

please find below some additional comments and integrations to my answers to the reviewers.

In the replies to referees' comments and in the revised manuscript, we have asserted that "The climatic impact associated to the deposition of BrC and dust on snowpack and its uncertainties were not reported in the IPCC 5th Assessment Report (AR5)". This is a truthful statement, however, it is not complete, since dust and brown carbon forcings in snow were actually discussed in "IPCC special report on the ocean and cryosphere in a changing climate" (SROCC) in the Chapter "Polar regions" (Meredith et al., 2019, pag. 247).

Specifically, in SROCC, the combined effect of black carbon and dust deposition on land-based snow is taken from Lawrence et al. (Journal of Climate, 2011), and is known with "medium confidence". The forcing from BrC is deduced from Lin et al. (ACP, 2014) (already cited in our work) and the related confidence is retained "low".

The revised manuscript has been updated including this information. The changes are highlighted in yellow in the text.